# How RL Unlocks the Aha Moment in Geometric Interleaved Reasoning

Xiangxiang Zhang [* 1]   Caijun Jia [* 2 3]   Siyuan Li [* 1 4]   Dingyu He [* 1]   Xiya Xiong [* 1]   Zheng Sun [2 3]   Honghao He [2 3]
Yuchen Wu [1]   Bihui Yu [2 3]   Linzhuang Sun [2 3]   Cheng Tan [† 4]   Jingxuan Wei [† 2 3 5]

## Abstract

Solving complex geometric problems inherently requires *interleaved reasoning*: a tight alternation between constructing diagrams and performing logical deductions. Although recent Multimodal Large Language Models (MLLMs) have demonstrated strong capabilities in visual generation and plotting, we identify a counterintuitive and underexplored phenomenon. Naively applying Supervised Fine-Tuning (SFT) on interleaved plot–solution data leads to a substantial degradation in reasoning performance compared to text-only baselines. We argue that this failure stems from a fundamental limitation of SFT, which primarily induces *distributional alignment*: the model learns to reproduce the surface format of interleaved plotting but fails to internalize the causal dependency between the generated plot and reasoning steps. To overcome this limitation, we propose Faire (**F**unctional **a**lignment for **i**nterleaved **re**asoning), a reinforcement learning framework that enforces three causal constraints to move beyond superficial imitation toward *functional alignment*. Extensive experiments show that Faire induces a qualitative shift in model behavior in which the plotting is effectively internalized, yielding competitive performance on challenging geometric reasoning benchmarks.

---

[*]Equal contribution  [1]ByteDance, Shanghai, China [2]Shenyang Institute of Computing Technology, Chinese Academy of Sciences, Shenyang, China [3]University of Chinese Academy of Science, Beijing, China [4]Westlake University, Hangzhou, China [5]Key Laboratory of Computing Power Network and Information Security, Ministry of Education, Shandong Computer Science Center (National Supercomputer Center in Jinan), China, Qilu University of Technology (Shandong Academy of Sciences), Shandong, China. Correspondence to: Cheng Tan <tancheng@pjlab.org.cn>, Jingxuan Wei <weijingxuan20@mails.ucas.edu.cn>.

*Proceedings of the 43rd International Conference on Machine Learning*, Seoul, South Korea. PMLR 306, 2026. Copyright 2026 by the author(s).

## 1. Introduction

Solving complex geometric problems is rarely a linear trajectory of purely textual deduction (Mouselinos et al., 2024; Xu et al., 2024). Instead, it requires interleaved reasoning, a dynamic interplay where experts iteratively construct visual diagrams to ground their logical derivations (Akter et al., 2024; Ning et al., 2025). Within this cognitive loop, the visual plot serves not merely as a static illustration, but as an essential cognitive scaffold: it exteriorizes working memory, makes latent spatial constraints explicit (Qi et al., 2025; Kumar et al., 2025), and actively steers the subsequent deductive path. With the rapid evolution of Multimodal Large Language Models (MLLMs), integrating such *thinking-in-plots* capabilities has emerged as a frontier pursuit (Lu et al., 2023; Zhuang et al., 2025). The ideal MLLM must therefore transcend the role of a passive calculator, evolving into a comprehensive solver (Li et al., 2025c; Wang et al., 2024a) that seamlessly interleaves code-based plotting with textual reasoning to achieve superior problem-solving accuracy.

However, existing training paradigms reveal a counterintuitive limitation (Zhang et al., 2023b; Wang et al., 2024b; Shao et al., 2024a). While Supervised Fine-Tuning (SFT) on high-quality interleaved plot–solution data is expected to improve reasoning by providing richer multimodal context (Gao et al., 2025a; Gu et al., 2025; Zheng et al., 2023), our empirical results show the opposite effect. As shown in Figure 1, naively applying SFT to such interleaved traces leads to a significant performance drop compared to text-only baselines. Rather than acting as a scaffold, the insertion of plotting steps disrupts the reasoning process by forcing the model to alternate between logical deduction and code generation. We argue that this failure stems from

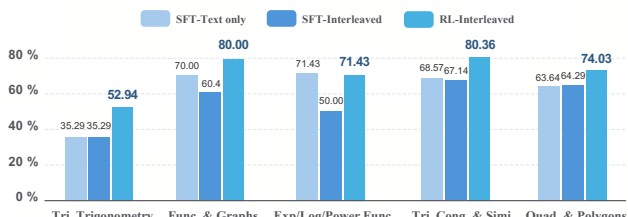

*Figure 1.* Illustration of challenges of *geometric interleaved reasoning* on several sub-tasks between SFT-Text only and SFT-Interleaved, which can only be tackled by RL post-training.

a fundamental limitation of SFT. SFT primarily induces distributional alignment by minimizing divergence from the training data, causing the model to imitate the surface form of interleaved reasoning, such as when to generate visual outputs, without internalizing the causal dependency that makes visual construction necessary for correct inference. As a result, visual generation is treated as a noisy pathway rather than an integral component of problem solving, which ultimately leads to the observed degradation in performance.

To bridge this gap in geometric reasoning, we posit that Reinforcement Learning (RL) is necessary to move beyond superficial *distributional alignment* toward *functional alignment*. We propose Faire (**F**unctional **a**lignment for **i**nterleaved **re**asoning), a RL–based framework that explicitly enforces functional alignment through preference optimization. Central to Faire is a tri-perspective verifier system that imposes strict causal constraints on the generation process. Specifically, an objective verifier validates geometric correctness through executable programs; a subjective verifier ensures visual perceptibility and interpretability via feedback from a vision–language model; and a semantic verifier enforces consistency between the generated reasoning, the constructed diagram, and the problem statement.

The functionally aligned reward mechanism mines the long-tail of the generation distribution, reinforcing those rare outlier trajectories where the diagram actively facilitates deduction. Faire not only reverses the negative transfer observed in SFT but also validates our theoretical hypothesis regarding functional alignment, ultimately establishing state-of-the-art performance on challenging geometric benchmarks where the model genuinely internalizes the interleaving thinking-in-plots paradigm.

## 2. Related Work

### 2.1. Multimodal Geometry Reasoning

Prior work on multimodal geometry problem solving can be broadly categorized into data-centric and perception-driven approaches. Representative efforts range from early neural solvers that align diagrams with textual representations (Zhang et al., 2023a), to instruction-tuned MLLMs and large-scale geometry datasets designed to enhance multimodal alignment and reasoning diversity (Gao et al., 2025b; Wu et al., 2023; Jaiswal et al., 2024; Huang et al., 2025b). Related studies further emphasize symbol-aware modeling to better ground geometric entities across text and vision (Ning et al., 2023). While these methods enhance perceptual grounding and data efficiency, their reasoning is weakly constrained, with implicit intermediate states.

In contrast, logic-centric approaches emphasize rigor and verifiability through neuro-symbolic or execution-based formulations, including language-guided symbolic sys-

tems (Trinh et al., 2024; Li et al., 2025b), formally structured representations and benchmarks (Zhang et al., 2024c; Wei et al., 2025b), executable code generation for geometry solving (Sharma et al., 2025), and formally verified supervision engines (Fu et al., 2025; Wu et al., 2025d). Interleaved paradigms that couple reasoning with explicit drawing actions have been proposed (Wu et al., 2025c). However, relying solely on supervised learning for such integration often leads to negative transfer, where models mimic the interaction without capturing its causal utility.

### 2.2. Interleaved Reasoning and Unified Generation

Recent multimodal research moves beyond static VQA toward *interleaved* reasoning traces and unified image-text generation. Representative efforts include interleaved chains of thought and progressive interleaved benchmarks (Gao et al., 2025a; Du et al., 2025), visual evidence chaining with region-level grounding (Shao et al., 2024a), and reflection-style interleaving for text-to-image generation (Huang et al., 2025a). In parallel, tool- and program-augmented paradigms translate perception into executable or inspectable intermediate states via tool orchestration, modular programs, and iterative search (Yang et al., 2023; Surís et al., 2023; Xu et al., 2025; Wang et al., 2025d; Ding et al., 2025). On the modeling side, unified architectures seek to handle interleaved understanding and generation within a single system, spanning unified I/O alignment (Koh et al., 2023), unified tokenization (Ge et al., 2024), single-transformer unification and autoregressive interleaved generation (Xie et al., 2025; Kou et al., 2024), as well as multimodal-prompted and interactive generation (Huang et al., 2025c).

To support this ecosystem, recent benchmarks evaluate open-ended interleaved outputs from multiple dimensions (Zhou et al., 2025; Liu et al., 2024b), while challenge-style testbeds provide a dynamic substrate for tracking rapidly evolving multimodal reasoning systems (Xu et al., 2025), and RL-based post-training further improves interleaved generation without requiring massive supervised trajectories (Nie et al., 2026). However, existing paradigms prioritize the form of interleaved generation over its functional utility, often resulting in the negative transfer that we observe in geometric tasks.

## 3. Method

### 3.1. Preliminaries

We formally define the task of *geometric interleaved reasoning*. Unlike standard text-only mathematical reasoning, where the solution is a continuous stream of logical tokens, interleaved reasoning requires the model to dynamically alternate between linguistic deduction and executable visual

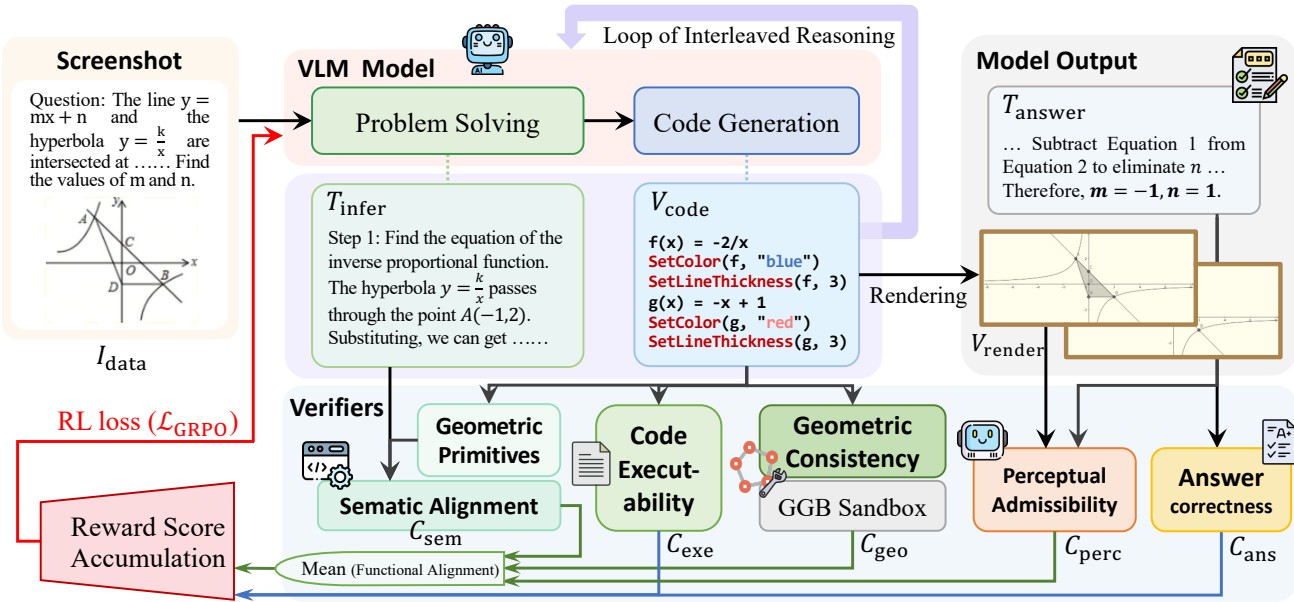

*Figure 2.* Illustration of the Faire framework. The model generates a reasoning trace and GeoGebra code from a geometry problem with a screenshot. As for reward designs, a gated reward enforces answer correctness $C_{ans}$ and code executability $C_{exe}$, then aggregates perceptual $C_{perc}$, semantic $C_{sem}$, and formal verification signals $C_{geo}$.

construction. Given a geometric problem $X$ that may include a problem statement and an initial diagram, the goal is to generate a solution trajectory $\tau$:

- **Text-only reasoning**: The trajectory is a homogeneous sequence of text tokens $\tau_{cot} = [t_1, t_2, ..., t_N]$, where each step is generated conditionally on the prior linguistic context, following $P(t_k|t_{<k}, X)$.

- **Interleaved reasoning**: The trajectory is a heterogeneous sequence of reasoning steps and plotting actions:

$$\tau_{int} = [(t_1, c_1), (t_2, c_2), ..., (t_k, c_K)]. \quad (1)$$

Here, $t_k$ represents a textual reasoning block, and $c_k$ represents an executable GeoGebra code block that generates a visual state $v_k = \text{Render}(c_k)$. The defining distinction between the two paradigms lies in the *conditional dependency*. In interleaved reasoning, the next textual step $t_{k+1}$ is conditioned not only on the preceding text, but also on the accumulated visual context:

$$P(t_{k+1}|t_{1:k}, v_{1:k}, X). \quad (2)$$

When properly aligned, the visual state $v_k$ serves as an external memory that clarifies latent constraints and facilitates subsequent deduction. Otherwise, the text-to-code transition introduces interference, and the generated diagram degrades into a nuisance signal that fragments the reasoning process.

### 3.2. Distributional Alignment Leads to Failure

To formally analyze the divergence between SFT and RL, we abstract the sequential trajectory into a probabilistic

graphical model. Let $T = \{t_1, ..., t_k\}$ denote the aggregate textual rationale, and $V = \{v_1, ..., v_k\}$ denote the aggregate visual artifacts generated throughout the reasoning process. The reasoning process is a joint distribution over the textual rationale $T$, the visual artifact $V$, and the final answer $Y$, conditioned on the problem context $X$.

SFT optimizes the policy $\pi_\theta$ to minimize the Kullback-Leibler (KL) divergence from the data distribution $\mathcal{D}$:

$$\mathcal{L}_{SFT}(\theta) = \mathbb{E}_{\tau \sim \mathcal{D}}[-\log \pi_\theta(T, V, Y|X)]. \quad (3)$$

The data distribution implies a causal structure where the expert generates $V$ specifically to simplify $Y : V \to Y$. However, SFT merely learns the conditional likelihoods:

$$\log \pi_\theta(\tau|X) = \underbrace{\log \pi_\theta(T_1|X)}_{\text{Premise}} + \underbrace{\log \pi_\theta(V|T_1, X)}_{\text{Plotting}}$$
$$+ \underbrace{\log \pi_\theta(T_2, Y|T_1, V, X)}_{\text{Deduction}} \quad (4)$$

We identify two fundamental issues under the SFT regime:

**(1) Superficial correlation** SFT minimizes the prediction error of $V$, *i.e.,* minimizing conditional entropy $H_\pi(V|T_1, X)$. The model learns to generate $V$ that looks like the training data (distributional alignment) but effectively treats $V$ and $Y$ as conditionally independent given strong textual priors:

$$I_{\text{data}}(V; Y|T_1, X) - I_{\pi_\theta}(V; Y|T_1, X) \gg 0. \quad (5)$$

Here, $I(\cdot; \cdot|\cdot)$ is conditional mutual information. The trained policy underestimates the dependency between $V$ and $Y$.

**(2) Context fragmentation** When $\pi_\theta$ generates a hallucinated scaffold, it acts as a noisy channel. Instead of reducing the uncertainty of the answer, the inclusion of $V$ introduces a distraction penalty. We formalize this as an inequality in conditional entropy:

$$H_{\pi_\theta}(Y|T_1, V, X) \geq H_{\pi_\theta}(Y|T_1, X) + \delta, \qquad (6)$$

where $\delta > 0$ represents the cognitive load of reconciling the conflict between the text context $T_1$ and the noisy diagram $V$. Thus, SFT interleaved data increases the entropy of the solution space rather than collapsing it.

### 3.3. Functional Alignment for Causal Dependency

To reverse the degradation caused by distributional alignment, we propose *functional alignment*. Unlike SFT, which treats the visual artifact $V$ as a supervised target, we model $V$ as a *latent causal mediator* in the reasoning graph $T \rightarrow V \rightarrow Y$. RL optimizes a policy that instantiates $V$ to serve as an information bridge between the textual premise $T$ and the answer $Y$. For the causal path $T \xrightarrow{\text{construct}} V \xrightarrow{\text{observe}} Y$ to be valid, the mediator $V$ must satisfy a set of necessary structural constraints. We define a Tri-perspective verification system $\mathcal{V}$ that formalizes these constraints, ensuring the structural integrity of the reasoning graph.

**Necessary conditions for causal mediation** We posit that $V$ is a functional mediator if and only if it satisfies three independent conditions: *Geometric Consistency* ($\mathcal{C}_{geo}$), *Perceptual Admissibility* ($\mathcal{C}_{perc}$), and *Semantic Alignment* ($\mathcal{C}_{sem}$).

**(1) Node Integrity** $\mathcal{C}_{geo}$: We require that the mediator $V$ exists as a well-defined mathematical object. Concretely, the generated code $c$ must be executable and not violate any axioms implied by the problem $X$. We formalize it via the *objective verifier*, which executes the code and checks against a set of hard constraints $\mathcal{K}_X$ derived from $X$:

$$\mathcal{C}_{geo}(c, X) = \mathbb{I}(c \in \Omega_{valid}) \cdot \prod_{k \in \mathcal{K}_X} \mathbb{I}(\text{Satisfies}(c, k)), \quad (7)$$

where $\Omega_{valid}$ denotes the space of syntactically valid programs, guaranteeing the ontological validity of the mediator.

**(2) Egress Validity** $\mathcal{C}_{perc}$: The information contained in $V$ must be transmittable to the solver. Even if $V$ is mathematically perfect, rendering artifacts (*e.g.,* occlusion, extreme scale, label overlap) act as channel noise, blocking the edge $V \rightarrow Y$. We formalize this via the *subjective verifier* using a MLLM as a judge function $J_{mllm}$:

$$\mathcal{C}_{perc}(v) = \mathbb{I}(J_{mllm}(v) > \tau_{perc}). \qquad (8)$$

Here, $\tau_{perc}$ is a threshold for visual clarity. Satisfying $\mathcal{C}_{perc}$ ensures that the channel capacity is sufficient.

**(3) Ingress Validity** $\mathcal{C}_{sem}$: The mediator $V$ must be causally downstream of the specific intent in $T$. A random valid figure (satisfying $\mathcal{C}_{geo}$ and $\mathcal{C}_{perc}$) that ignores the current reasoning plan implies a broken edge $T \nrightarrow V$ (*i.e.,* mutual information $I(T; V) \approx 0$).We formalize this via the *semantic verifier*. Let $\phi(c)$ be the set of geometric primitives extracted from code, and $\psi(T)$ be the set of geometric intents extracted from text. We require logical entailment:

$$\mathcal{C}_{sem}(c, T) = \mathbb{I}(\psi(T) \subseteq \phi(c)). \qquad (9)$$

This condition enforces that the diagram is not just a valid figure, but the specific figure necessitated by the textual logic, securing the ingress edge of the mediation.

**Completeness of Verification** The proposed system is sufficient to establish functional alignment because it covers all components of the local causal graph: $\mathcal{C}_{sem}$ validates the input edge ($T \rightarrow V$), $\mathcal{C}_{geo}$ validates the node integrity ($V$), and $\mathcal{C}_{perc}$ validates the output edge ($V \rightarrow Y$):

$$T \underset{\mathcal{C}_{sem}}{\xrightarrow{\text{construct}}} \underbrace{V}_{\mathcal{C}_{geo}} \underset{\mathcal{C}_{perc}}{\xrightarrow{\text{observe}}} Y. \qquad (10)$$

**Optimization** We integrate the proposed constraints into a unified dense reward function:

$$R(V, Y) = \mathbb{I}(Y = Y^*) + \beta(\mathcal{C}_{geo} + \mathcal{C}_{perc} + \mathcal{C}_{sem}). \quad (11)$$

Using GRPO (Shao et al., 2024b), we update the policy to maximize the expectation of this causally-grounded reward:

$$\mathcal{L}_{GRPO} = \mathbb{E}_{\tau \sim \pi_\theta} \left[ \frac{\pi_\theta(\tau)}{\pi_{old}(\tau)} \hat{A}(\tau) \right] - \beta D_{KL}(\pi_\theta || \pi_{ref}). \tag{12}$$

By explicitly rewarding these three conditions, Faire forces the model to internalize the causal structure of interleaved reasoning, ensuring that every generated plot is functionally instrumental in deriving the final solution.

## 4. Dataset

We introduce **Faire-Bench**, a benchmark for *geometric interleaved reasoning* built with a synthesis-verification pipeline in Figure 3. Each instance couples stepwise deduction with an executable program whose execution reconstructs the geometric state that the next deduction is meant to rely on.

### 4.1. Dataset Construction

We construct a K12 dataset of aligned solution traces and executable GeoGebra scripts via constrained synthesis and a verification funnel that yields *All-Pass* samples.

**Taxonomy-driven selection** We filter a large pool of K12 problems: A lightweight taxonomy covers (i) function and analytic-geometry tasks that depend on axes, curves, and

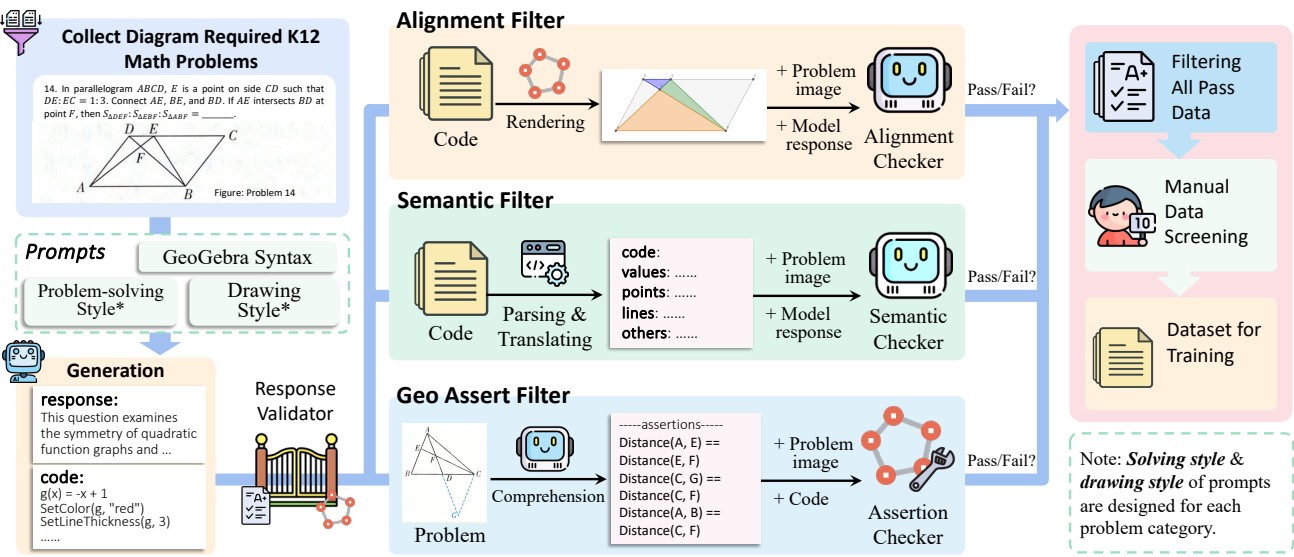

*Figure 3.* The data construction pipeline employing a tri-perspective verification mechanism—Visual Alignment, Semantic Consistency, and Geometric Assertion—to curate rigorous interleaved geometric reasoning samples.

key points (*e.g.,* intercepts, vertices, intersections, tangency points), and (ii) Euclidean construction tasks defined by primitives and relations (*e.g.,* collinearity, perpendicularity, parallelism, tangency) with readable annotations.

**Constrained synthesis** For each selected problem, Gemini 2.5 generates a stepwise solution trace and a GeoGebra script. Prompts enforce three minimal constraints: (1) **Syntax**, the script is standalone and free of natural-language artifacts; (2) **Grounding**, reasoning steps explicitly reference the constructed objects; (3) **Drafting**, the diagram follows category-specific conventions (*e.g.,* axes and key-point labels for plots, construction order for Euclidean tasks).

**Hard filtering** We discard incorrect candidates, then execute code in GeoGebra to ensure it constructs a valid geometric state and produces a non-degenerate render $I_{\text{render}}$.

**Verification funnel** We apply three complementary filters that target common failure modes: (1) **Alignment Filter** compares $I_{\text{render}}$ against the problem diagram $I_{\text{prob}}$ (with response as context) to reject missing elements, disconnected auxiliary strokes, off-canvas renders, and severe label clutter. (2) **Semantic Filter** parses code into $T_{\text{IR}}$ and verifies symbolic consistency between response and $T_{\text{IR}}$, preventing intent-shifted constructions that remain visually plausible. (3) **Geo Assert Filter** synthesizes geometric assertions from the problem specification and evaluates them in the GeoGebra kernel, providing a deterministic truth verdict for key relations. Only candidates that pass all filters enter expert review; the survivors form the final dataset.

### 4.2. Dataset analysis

**Split consistency and visual context** Our corpus contains 7,989 instances, partitioned into an SFT (4,643)/RL

(2,321)/evaluation(1,025) splits, summarized in Table 1. All three splits exhibit highly consistent distributions in category composition and difficulty: *Plane Geometry* accounts for about 52–55%, followed by *Function* (27–29%) and *Analytic Geometry* (18–20%), while hard problems consistently comprise about 54% of each split. A distinctive property of the dataset is its substantial multi-image coverage. Across splits, 38–44% of instances contain at least two images.

*Table 1.* Consistent split-level statistics of the corpus.

|  | SFT | RL | Eval |
|---|---|---|---|
| #Instances | 4643 | 2321 | 1025 |
| Plane geometry | 51.5% | 51.6% | 54.7% |
| Function | 29.4% | 28.3% | 27.4% |
| Analytic geometry | 19.1% | 20.1% | 17.9% |
| Hard difficulty | 54.4% | 54.8% | 54.1% |
| ≥2 images | 39.6% | 38.4% | 43.6% |

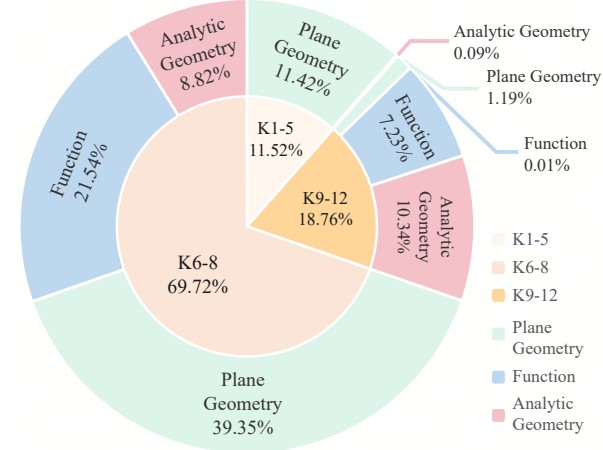

*Figure 4.* Stage and category distribution. Inner ring shows educational stages; outer ring shows category shares within each stage.

**Educational stages and categories** Figure 4 shows the interaction between educational stages and geometric categories. Middle-school content (K6–8) dominates the corpus (about 69%), while K9–12 contributes 17–20% and K0–5 contributes 11–14%. This skew reflects the diagram-required selection criterion: K6–8 is the stage where students transition from arithmetic to explicit geometric reasoning, making diagrammatic construction central rather than auxiliary.

**Fine-grained skill coverage** At the sub-category level, the corpus concentrates on triangle congruence and similarity (27.8%), function graphs (25.6%), and polygon reasoning (14.7%), followed by line–coordinate geometry (9.1%) and conic sections with linear relations (6.2%). These skills are inherently diagram-heavy and require precise constructions with faithful alignment between textual reasoning and visual evidence, reinforcing the dataset's suitability for studying when construction becomes a functional part of reasoning.

## 5. Experiment

### 5.1. Experimental Setup

We evaluate Faire against a broad set of strong multimodal baselines, including both proprietary and open-weight MLLMs. The proprietary models include GPT-4o (Hurst et al., 2024), GPT-5.1 (OpenAI, 2025b), GPT-5.2 (OpenAI, 2026), and Gemini-2.5-Pro (Comanici et al., 2025). For open-weight baselines, we consider GLM-4.1V-9B (Zeng et al., 2025), Gemma3-12B (Team, 2025a), InternVL3.5-8B (Wang et al., 2025c), Kimi-VL-A3B (Team, 2025b), Qwen2.5-VL-7B (Team, 2025c), and Qwen3-VL-8B (Bai et al., 2025). Unless otherwise specified, all models follow the same prompting protocol and evaluation budget.

**Supervised initialization** We initialize Faire from Qwen3-VL-8B and apply SFT with bfloat16 precision and SDPA attention. SFT runs on a single 8-GPU node with a visual token cap of 2,048 and a maximum sequence length of 10,000 to accommodate interleaved text–code contexts. We train for 2 epochs with learning rate $5 \times 10^{-6}$ and warmup ratio 0.05, and use DeepSpeed ZeRO-2 for memory optimization.

**Reinforcement learning** Starting from the SFT checkpoint, we apply GRPO to refine interleaved trajectories under our verification signals. We accelerate sampling with vLLM and generate 7 candidates per prompt with temperature 0.9.

### 5.2. Main Results

Table 2 reports both answer accuracy and our tri-perspective verification, which operationalizes the difference between *distributional alignment* (matching the interleaving format) and *functional alignment* (constructing states that subsequent deductions can verifiably rely on). Faire achieves the strongest functional alignment by a wide margin: Avg

reaches 45.37, while the best proprietary baselines remain below 25 (Gemini-2.5-Pro (Comanici et al., 2025) at 19.86; GPT-5.2 (OpenAI, 2026) at 24.62). The lead is structural rather than cosmetic: Faire improves all three verification views at once, with Parser at 37.27, Code at 60.39, and Judge at 38.44. These results support the following observations.

**Accuracy leader does not imply interleaved reasoning.** Gemini-2.5-Pro (Comanici et al., 2025) attains the highest answer accuracy, yet its constructions fail to carry the proof burden: formal validity remains at 22.51 and the overall verification average stays at 19.86. This discrepancy suggests that the model likely relies on textual shortcuts or internal knowledge to solve problems, treating the plotting step as a detached ritual rather than a computational tool. In contrast, Faire pairs strong answer accuracy with substantially stronger formal grounding, nearly tripling formal validity (60.39 versus 22.51), indicating that it reasons *through* verifiable constructions rather than independently of them.

**Surface mimicry vs. causal grounding.** GPT-5.2 (OpenAI, 2026) often produces diagrams that visually resemble the reference, reflected in high SSIM and LPIPS scores, yet its semantic alignment collapses, with Parser dropping to 7.02. This pattern exemplifies distributional alignment: the output matches appearance statistics while failing to instantiate the relations required by the reasoning. Faire avoids this pitfall by prioritizing verifiable relations over surface resemblance, achieving much stronger semantic and formal consistency.

**RL quantifies the Aha Moment.** Relative to its supervised initializer Qwen3-VL-8B (Bai et al., 2025), Faire raises the overall verification score from 14.85 to 45.37. The largest gains appear precisely on the signals that encode functional coupling: Parser improves from 8.62 to 37.27, and Code from 25.47 to 60.39. This quantitative leap captures the *Aha Moment*: the visual generation transitions from a fragile, formatting artifact into a robust, load-bearing scaffold. The model has not just optimized a metric; it has internalized the tool, learning that correct plotting is the necessary causal antecedent to correct reasoning.

### 5.3. The Aha Moment: Entropy Shifts under RL

Figure 5 analyzes the top-100 tokens whose entropy increases most from SFT to RL. Here, higher entropy does not indicate randomness. Instead, it reflects a shift away from low-entropy, template-driven generation toward deliberate computation. The largest entropy increase appears on *visual drawing tokens* (23.38%), showing that diagram construction is no longer executed as a fixed routine. Under RL, drawing becomes an active decision process shaped by verifier feedback, rather than a replay of memorized patterns. Notably, *numeric* (21.36%) and *code* (17.63%) tokens together account for nearly 40% of the entropy shift. Their entropy increase indicates that the model moves from guess-

*Table 2.* Main results on interleaved geometry reasoning. Acc: answer accuracy. Verification uses tri-perspective scores: Parser ($v_{\text{sem}}$), Code ($v_{\text{form}}$), Judge ($v_{\text{vis}}$). Similarity metrics (BLEU/ROUGE-L/chrF; PSNR/SSIM/LPIPS) report surface matching to references.

| Model | Acc | Code Similarity | | | Image Similarity | | | Verification Scores | | | |
|---|---|---|---|---|---|---|---|---|---|---|---|
| | | BLEU | ROUGE-L | chrF | PSNR | SSIM | LPIPS | Parser | Code | Judge | Avg |
| Gemma3-12B (Team, 2025a) | 20.97 | 7.43 | 25.35 | 33.57 | 0.83 | 3.60 | 97.56 | 1.97 | 4.13 | 0.88 | 2.33 |
| Kimi-VL-A3B (Team, 2025b) | 23.22 | 5.00 | 15.21 | 27.18 | 1.45 | 6.53 | 95.70 | 2.61 | 11.94 | 6.82 | 7.12 |
| InternVL3.5-8B (Wang et al., 2025c) | 45.56 | 8.57 | 27.49 | 32.81 | 4.77 | 21.33 | 85.76 | 5.00 | 17.84 | 5.20 | 9.35 |
| Qwen2.5-VL-7B (Team, 2025c) | 26.34 | 3.97 | 11.60 | 16.66 | 0.60 | 2.61 | 98.27 | 5.16 | 21.83 | 3.31 | 10.10 |
| GLM-4.1V-9B (Zeng et al., 2025) | 63.41 | 5.86 | 15.70 | 23.97 | 0.67 | 2.98 | 98.05 | 7.43 | 22.39 | 7.04 | 12.29 |
| Qwen3-VL-8B (Bai et al., 2025) | 59.71 | 16.72 | 36.71 | 54.96 | 6.32 | 27.02 | 81.50 | 8.62 | 25.47 | 10.47 | 14.85 |
| GPT-4o (Hurst et al., 2024) | 25.56 | 9.14 | 27.13 | 36.01 | 2.16 | 9.28 | 93.50 | 4.62 | 8.92 | 5.27 | 6.27 |
| GPT-5.1 (OpenAI, 2025b) | 56.58 | 11.85 | 32.78 | 47.08 | 7.39 | 32.50 | 76.50 | 10.34 | 25.85 | 18.24 | 18.14 |
| Gemini-2.5-Pro (Comanici et al., 2025) | **78.24** | 20.20 | 38.62 | 54.70 | 4.82 | 21.67 | 85.15 | 15.90 | 22.51 | 15.45 | 19.86 |
| GPT-5.2 (OpenAI, 2026) | 68.78 | 2.32 | 9.77 | 44.32 | 9.85 | 43.79 | 69.23 | 7.02 | 36.59 | 30.24 | 24.62 |
| **Faire** (Ours) | 74.82 | **25.06** | **46.75** | **55.57** | **14.79** | **65.10** | **52.76** | **37.27** | **60.39** | **38.44** | **45.37** |

ing numerically plausible values to explicitly computing and validating them, a hallmark of reliable geometric reasoning. Finally, *reasoning-chain tokens* (16.60%) also exhibit elevated entropy, suggesting that the model no longer commits early to a fixed solution path, but allows constructed diagrams to influence subsequent deductions. We view it as the behavioral signature of the **Aha moment**, where construction becomes functionally integrated into reasoning.

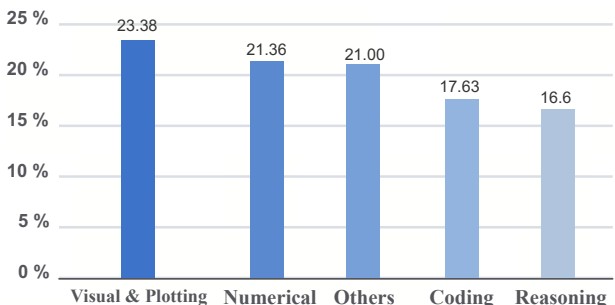

*Figure 5.* Distribution of top-100 entropy-increased tokens after RL. Tokens are grouped by semantic function.

### 5.4. The Aha Moment: When Drawing Hurts No More

Figure 6-7 illustrates the failure mode behind the SFT paradox and the behavioral shift induced by Faire. With interleaved SFT, the model can mimic the *format* of alternating reasoning and drawing, yet the construction is only loosely tied to the proof: the diagram highlights the claimed congruence but does not faithfully implement the intended visual hierarchy, so the "image step" cannot reliably support the "reasoning step". In contrast, Faire produces a construction that is operationally consistent with the proof: auxiliary points and segments are instantiated in the right dependency order, and the rendered figure functions as *evidence* rather than decoration. Under SFT, the act of drawing is a distractor that consumes context window with mismatched information. Under Faire, drawing becomes a valid working

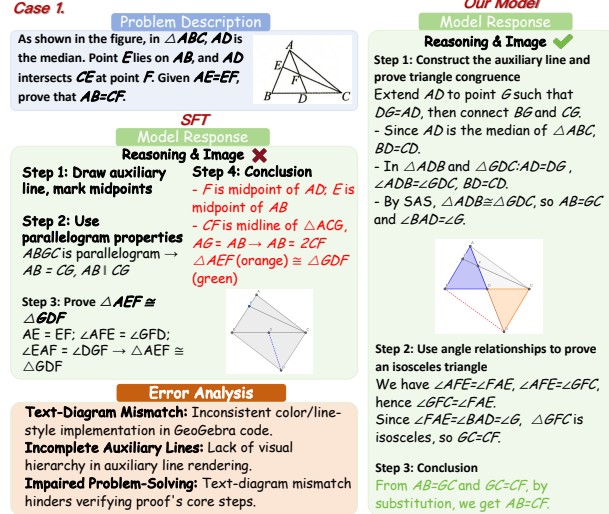

*Figure 6.* RL vs. SFT Text–Diagram Alignment in a Geometry problem, where RL model learns when to draw during reasoning.

state that the subsequent deduction can rely on. It is the Aha moment we target—drawing becomes a working state the next deduction can depend on—matching the large gains in Table 2 and the RL-driven entropy reallocation in Figure 5.

### 5.5. The Aha Moment: Structure over Relations

Table 3 evaluates Faire on GenExam (Wang et al., 2025e), which separates *structural correctness* (Math-Str) from *relational correctness* (Math-Rel), testing whether the RL-induced behavior we target in geometric interleaved reasoning transfers beyond geometry: not just describing relations, but maintaining a coherent underlying structure.

The contrast is sharp: several strong baselines achieve high Math-Rel (*e.g.,* GPT-Image-1 (Hurst et al., 2024) at 52.0) while scoring 0.0 on Math-Str, indicating that relation-level plausibility can be produced without committing to a valid

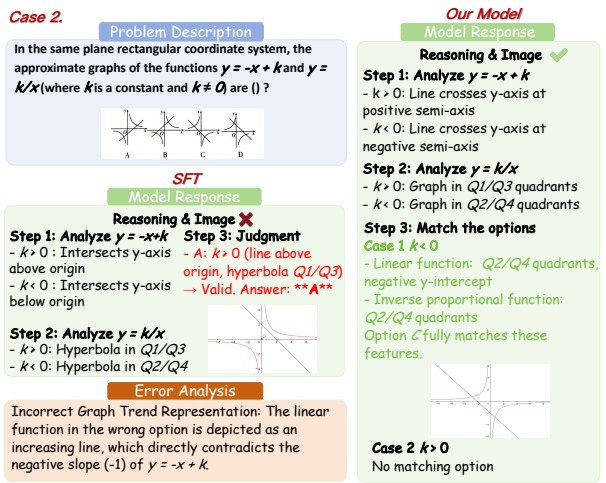

*Figure 7.* RL vs. SFT in Function Graph. The SFT model draws incorrect quadrant distributions with text rationales, while the RL model draws an accurate image to infer the correct answer.

*Table 3.* Results on GenExam. Math-Str: structural correctness; Math-Rel: relational correctness.

| Model | Math-Rel | Math-Str |
|---|---|---|
| Emu3 (Wang et al., 2024c) | 11.3 | 0.0 |
| Janus-Pro (Chen et al., 2025) | 13.7 | 0.0 |
| Qwen-Image (Wu et al., 2025b) | 18.9 | 0.0 |
| Imagen-4-Ultra (Saharia et al., 2022) | 35.9 | 2.6 |
| Gemini-2.5-Image (Comanici et al., 2025) | 43.1 | 0.7 |
| GPT-Image-1 (Hurst et al., 2024) | 52.0 | 8.0 |
| **Faire** (Ours) | **52.3** | **9.3** |

construction. Faire breaks this pattern, reaching the best Math-Str (9.3) while also matching the best Math-Rel (52.3). This supports our central claim about the *Aha Moment*: RL shifts the model from imitating relational outcomes to prioritizing structure that can actually support reasoning.

### 5.6. Ablation Study

We ablate Faire to validate two claims: (i) interleaved supervision under SFT can induce a structural failure mode (the *SFT paradox*), and (ii) reinforcement learning is required to recover *functional alignment* between constructed geometric states and subsequent deductions.

**Reward completeness is essential.** Table 4 reports a leave-one-out ablation of the Faire reward. Optimizing with an executability-only signal yields only modest improvements, indicating that RL alone is insufficient. In contrast, removing any single verifier consistently degrades both accuracy and Draw Avg, showing that no proxy is adequate on its own and the three signals are genuinely complementary.

**Interleaving fails under SFT but succeeds under RL.** Table 5 isolates the paradox. Interleaving under SFT reduces accuracy relative to a text-only pipeline, suggesting that the

*Table 4.* Reward ablation (leave-one-out). Executability-only RL provides limited gains; Faire requires all verifiers to achieve functional alignment.

| Model | Acc | Draw Avg |
|---|---|---|
| Faire (Full) | **74.83** | **45.37** |
| Faire w/o Alignment | 66.53 | 38.31 |
| Faire w/o Semantic | 66.24 | 39.37 |
| Faire w/o Geo Assert | 65.88 | 37.15 |
| RL (Exec-only) | 64.29 | 35.65 |
| SFT (Interleaved) | 62.48 | 35.23 |

model can imitate alternation without internalizing its causal role. After RL, the same interleaved pipeline reverses this degradation and outperforms its text-only counterpart, drawing shifts from a distraction to a usable intermediate state once the objective enforces state-to-reasoning coupling.

*Table 5.* Ablation on Training Paradigms. Comparing the interplay between drawing and solving. Note that while drawing initially hurts SFT accuracy, it boosts performance post-RL.

| Paradigms | SFT | | RL | |
| | Acc (%) | Draw | Acc (%) | Draw |
|---|---|---|---|---|
| Text-only | 68.13 | - | 71.21 | - |
| Interleaved | 62.48 | 35.23 | **74.83** | **45.37** |

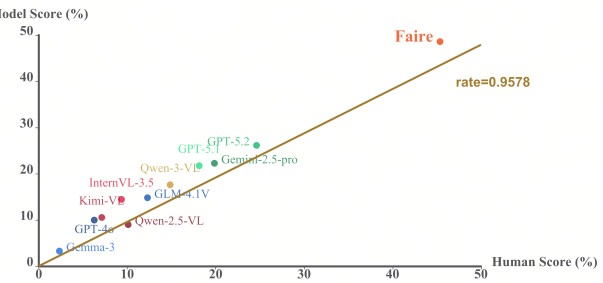

*Figure 8.* Verifier score vs. human rating ($r = 0.9578$).

### 5.7. Consistency with Human Judgments

Figure 8 shows a clear spatial separation. Most existing MLLMs, including GPT-4o (Hurst et al., 2024) and Gemini-2.5-Pro (Comanici et al., 2025), cluster in the lower-left region with both low verifier scores and low human ratings. In contrast, Faire is the only model that occupies the top-right corner, achieving simultaneously high automatic scores and high human preference. The near-perfect correlation ($r = 0.9578$) demonstrates a strong alignment between automated verification and human judgment. Despite being optimized via reinforcement learning, verifier improvements translate monotonically into human judgment, indicating that the reward cannot be exploited by superficial artifacts. This confirms that our tri-perspective verification captures genuine diagram usability rather than proxy signals. Overall, Faire does not merely improve scores but establishes

*Table 6.* Results on GGBench. Stage-wise interleaved evaluation (higher is better) and image similarity (lower LPIPS is better).

| Model | Stage-wise scores | | | Image similarity | | | Overall |
|---|---|---|---|---|---|---|---|
| | Planning (VLM-T)↑ | Mid (VLM-I-Mid)↑ | Final (VLM-I-Res)↑ | LPIPS×$10^{-2}$ ↓ | PSNR↑ | SSIM×$10^{-2}$ ↑ | VLM-I↑ |
| *End-to-end UMMs* | | | | | | | |
| Qwen-Image (Wu et al., 2025b) | — | — | 22.75 | 56.39 | 58.23 | 48.06 | 22.75 |
| Janus (Wu et al., 2025a) | 33.85 | 21.69 | 19.76 | 57.74 | 57.76 | 60.97 | 20.73 |
| Nano Banana (Comanici et al., 2025) | 58.54 | 44.83 | 22.81 | 51.85 | 64.53 | 59.51 | 33.82 |
| GPT-image-1 (Hurst et al., 2024) | — | — | 39.11 | 50.67 | 16.95 | 64.37 | 39.11 |
| *LLMs/LRMs* | | | | | | | |
| GPT-4o (Hurst et al., 2024) | 59.73 | 26.19 | 2.66 | 95.43 | 5.45 | 5.69 | 14.43 |
| GLM-4.5V (Hong et al., 2025) | 53.32 | 25.63 | 5.02 | 52.91 | 12.19 | 12.94 | 15.33 |
| Qwen3-14B (Yang et al., 2025) | 58.65 | 39.30 | 12.97 | 78.81 | 23.92 | 24.81 | 26.13 |
| Gemini 2.5 Pro (Comanici et al., 2025) | 38.50 | 37.41 | 15.80 | 68.39 | 37.17 | 39.73 | 26.61 |
| DeepSeek-R1 (Liu et al., 2024a) | 61.16 | 62.42 | 20.48 | 66.06 | 37.94 | 37.59 | 41.45 |
| GPT-4 (Achiam et al., 2023) | 55.66 | 50.39 | 20.30 | 67.35 | 35.26 | 38.31 | 33.04 |
| Qwen3-VL (Yang et al., 2025) | 56.40 | 49.55 | 23.94 | 39.40 | 52.33 | 58.71 | 36.74 |
| DeepSeek-V3.1 (Liu et al., 2024a) | 60.24 | 73.13 | 26.41 | 57.21 | 48.33 | 50.12 | 49.77 |
| Claude Sonnet 4.5 (Anthropic, 2025) | 61.19 | 77.92 | 30.29 | 52.22 | 51.74 | 50.52 | 54.11 |
| GPT-5 (OpenAI, 2025a) | **62.01** | 76.79 | **37.36** | 49.65 | **54.80** | 59.49 | 57.08 |
| **Faire** (Ours) | 60.13 | **89.11** | 37.14 | **33.31** | 20.36 | **75.57** | **63.13** |

*Table 7.* RL vs. Best-of-$N$ on the SFT checkpoint. Best-of-$N$ uses our tri-perspective verifiers as the selection criterion.

| Method | Acc (%) | Draw Avg |
|---|---|---|
| SFT (Interleaved) | 62.48 | 35.23 |
| SFT BoN ($N = 5$) | 66.14 | 36.18 |
| SFT BoN ($N = 10$) | 69.73 | 37.92 |
| **Faire** (Ours) | **74.83** | **45.37** |

a distinct regime where automated verification and human evaluation are tightly aligned, validating the robustness of our reward design for interleaved geometric reasoning.

### 5.8. Beyond Standard RLVR

Faire is not a straightforward application of standard RLVR to a new domain. In interleaved geometry, SFT on multimodal traces can cause negative transfer: the model imitates the surface alternation between reasoning and drawing without learning their causal dependency. RL must therefore first overcome this SFT paradox before improving reasoning. To test whether Faire's gains come from inference-time selection, we apply Best-of-$N$ sampling to the SFT checkpoint with the same tri-perspective verifiers. As shown in Table 7, SFT accuracy rises from 62.48 to 69.73 at $N = 10$, yet remains below Faire's 74.83. The drawing score gap is even larger, with SFT BoN reaching 37.92 versus Faire's 45.37. These results suggest that Best-of-$N$ can select better SFT traces, but cannot repair the underlying construction–reasoning coupling.

### 5.9. Results on GGBench

Table 6 reports stage-wise evaluation on GGBench (Wei et al., 2025a), including *Planning* (VLM-T), *Middle Process* (VLM-I-Mid), *Final Result* (VLM-I-Res), pixel-level similarity, and *Overall Interleaved Score* (VLM-I). This benchmark examines whether a model can reason through

constructed geometric states rather than only produce plausible final answers. Faire achieves the best VLM-I of 63.13, despite being much smaller than proprietary systems such as GPT-5 and Claude Sonnet 4.5. Its advantage mainly comes from the *Middle Process* score, where Faire reaches 89.11 while all baselines remain below 78, indicating stronger step-by-step geometric state grounding. Pixel-level metrics show a similar trend: Faire obtains the highest SSIM despite relatively low PSNR, suggesting that it better preserves geometric structure and relations rather than merely optimizing visual smoothness. Moreover, GGBench demonstrates that functional alignment is more important than model scale or surface-level image quality for reliable interleaved geometric reasoning.

## 6. Conclusion

Geometric problem solving often requires *interleaved reasoning*, where diagram construction and logical deduction must support each other step by step. We surface a counterintuitive *SFT paradox*: supervised fine-tuning on interleaved traces can reduce solving accuracy, because it fits the *distribution* of alternation while leaving the construction–reasoning dependency under-optimized. We address this gap with Faire, which enforces *functional alignment* via reinforcement learning with gated rewards and post-generation verification from complementary visual, semantic, and formal verifiers. Across benchmarks and categories, Faire turns construction into a dependable intermediate state for the next deduction, capturing the Aha Moment that interleaving is meant to enable. Importantly, the verifier-driven scores track human judgments closely, supporting that the learned gains reflect better grounding rather than reward hacking. We hope this work encourages verifiable interleaved reasoning beyond geometry, and motivates stronger assertion synthesis with executable tools.

## Acknowledgments

The authors thank anonymous reviewers for their valuable comments and insightful feedback. The authors also thank Shenyang Institute of Computing Technology, Chinese Academy of Sciences, and ByteDance for their support. This work was supported by the ECAP project development agreement between ByteDance and the Shenyang Institute of Computing Technology, Chinese Academy of Sciences. This work was done by Caijun Jia, Siyuan Li, Dingyu He, Xiya Xiong, Zheng Sun, Honghao He, Lingzhuang Sun during their research internship at ByteDance. The trained models and training datasets will not be available to public because of commercial and security constrains.

## Impact Statement

This work aims to advance the field of machine learning by studying how multimodal models can perform reliable interleaved reasoning with executable geometric constructions. Our primary contributions are methodological and empirical, focusing on training objectives, verification mechanisms, and evaluation protocols, rather than on deploying models in real-world decision-making systems.

We do not anticipate direct negative societal impacts from this work. The proposed framework emphasizes verifiability, consistency, and transparency in model behavior, which are generally aligned with broader goals of improving the reliability and interpretability of AI systems. While more capable reasoning and tool-use models could, in principle, be misused in downstream applications, such risks are not specific to our approach and are orthogonal to the core technical contributions presented here.

Overall, we believe this work contributes positively to foundational research in machine learning, and we see no ethical concerns that require special mitigation beyond standard responsible research practices.

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

# A. Evaluation Metrics and Protocols

## A.1. Evaluation metrics

We evaluate multimodal geometry solving along two axes: *solution rigor* (answer correctness) and *diagram quality* (whether the generated GeoGebra construction is usable, aligned with the intended reasoning, and geometrically valid). Since geometric constructions admit many equivalent realizations, we report both intention-oriented verification scores and surface-level similarity scores. All automatic judges follow fixed prompts and deterministic execution rules for reproducibility; detailed protocols appear in Appendix A.

**Answer accuracy.** We extract the final answer $\hat{a}$ from the model output and compare it with the reference answer $a^\star$ under mathematical equivalence. Accuracy is binary: 1 if correct (or equivalent) and 0 otherwise.

**Drawing score.** To assess drawing capability beyond appearance matching, we use a tri-perspective evaluation that targets three complementary failure modes in executable geometry: perceptually unusable renders, intent-shifted constructions, and formally invalid relations. For each sample, we compute three normalized scores in $[0, 1]$ and report their mean as the final drawing score:

$$S_{\text{draw}} = \frac{1}{3}\Big(s_{\text{vis}} + s_{\text{sem}} + s_{\text{form}}\Big). \tag{13}$$

Here $s_{\text{vis}}$ comes from the **Visual Judge** (VLM-Judge), which evaluates perceptual usability by judging the rendered image $I_{\text{render}}$ against the problem diagram $I_{\text{prob}}$ with the solution trace as context. $s_{\text{sem}}$ comes from the **Semantic Judge** (Parser-Judge), which parses the program into an intermediate representation $T_{\text{IR}}$ and checks consistency between the reasoning trace and $T_{\text{IR}}$. $s_{\text{form}}$ comes from the **Formal Judge** (Assert-Judge), which executes synthesized geometric assertions in the GeoGebra kernel and scores the assertion pass rate. We also report the three sub-scores separately to diagnose failure patterns such as "looks right but violates constraints" or "formally correct but unreadable".

**Surface similarity metrics.** For completeness, we additionally report objective similarity scores that quantify surface-level proximity to references. *Code similarity* measures how close the generated GeoGebra script is to a reference script using BLEU, ROUGE-L, chrF, Edit Distance, and RUBY (higher is better except Edit Distance). *Image similarity* compares $I_{\text{render}}$ with the reference diagram using PSNR, SSIM, and LPIPS (higher is better except LPIPS). These similarity metrics are not reliable proxies for geometric correctness under an open-ended solution space, but they help characterize formatting and appearance alignment.

## A.2. Tri-perspective drawing evaluation

**Visual Judge (VLM-Judge).** We execute the generated GeoGebra program to obtain a rendered image $I_{\text{render}}$. A vision-language judge receives $(I_{\text{prob}}, I_{\text{render}}, T_{\text{infer}})$ and outputs a normalized score $s_{\text{vis}} \in [0, 1]$. The rubric checks (i) element completeness within the viewport, (ii) connectivity of auxiliary constructions, (iii) legibility, and (iv) structural consistency with the intended diagram. Rendering failure yields $s_{\text{vis}} = 0$.

**Semantic Judge (Parser-Judge).** We parse the generated program into a structured intermediate representation $T_{\text{IR}}$ that lists primitives and relations together with attributes and dependencies. A judge checks whether the reasoning trace $T_{\text{infer}}$ is consistent with $T_{\text{IR}}$ and outputs $s_{\text{sem}} \in [0, 1]$. Parsing failure or missing required entities yields $s_{\text{sem}} = 0$.

**Formal Judge (Assert-Judge).** We synthesize a set of executable assertions from the problem specification and evaluate them in the GeoGebra kernel on the constructed state. We define the formal score as a weighted pass rate:

$$s_{\text{form}} = \frac{\sum_j w_j \cdot \mathbb{1}[\text{assertion}_j \text{ passes}]}{\sum_j w_j}, \tag{14}$$

where $w_j$ assigns higher weight to key assertions directly implied by the problem. Optionally, key-assertion failure triggers $s_{\text{form}} = 0$ to prevent partial-credit artifacts.

## A.3. Code similarity metrics

We treat GeoGebra scripts as structured text and compute surface similarity against a reference script. We tokenize by commands, identifiers, numbers, and delimiters. We report BLEU, ROUGE-L, chrF, Edit Distance, and RUBY. RUBY

*Table 8.* **Comparison with existing multimodal math benchmarks.** We contrast capability coverage (solving, drawing, interleaved solving) and supervision modality (text, image, code).

| Benchmark | Solving | Drawing | Interleaved | Text | Image | Code |
|---|---|---|---|---|---|---|
| MathVista (Lu et al., 2023) | ✓ | ✗ | ✗ | ✓ | ✓ | ✗ |
| MathVerse (Zhang et al., 2024b) | ✓ | ✗ | ✗ | ✓ | ✓ | ✗ |
| MM-MATH (Sun et al., 2024) | ✓ | ✗ | ✗ | ✓ | ✓ | ✗ |
| MathScape (Zhou et al., 2024) | ✓ | ✗ | ✗ | ✓ | ✓ | ✗ |
| GeoEval (Zhang et al., 2024a) | ✓ | ✗ | ✗ | ✓ | ✓ | ✗ |
| WE-MATH (Qiao et al., 2024) | ✓ | ✗ | ✗ | ✓ | ✓ | ✗ |
| MMSciBench (Ye et al., 2025) | ✓ | ✗ | ✗ | ✓ | ✓ | ✗ |
| MATH2VISUAL (Wang et al., 2025a) | ✓ | ✓ | ✗ | ✓ | ✓ | ✗ |
| PolyMath (Gupta et al., 2024) | ✓ | ✗ | ✗ | ✓ | ✓ | ✗ |
| SOLIDGEO (Wang et al., 2025b) | ✓ | ✗ | ✗ | ✓ | ✓ | ✗ |
| GGBench (Wei et al., 2025a) | ✗ | ✓ | ✗ | ✓ | ✓ | ✓ |
| **Faire-Bench** (Ours) | ✓ | ✓ | ✓ | ✓ | ✓ | ✓ |

applies lightweight normalization (*e.g.,* whitespace removal and numeric canonicalization) before matching and weights key construction commands more heavily.

### A.4. Image similarity metrics

We compare $I_{\text{render}}$ with a reference diagram using PSNR, SSIM, and LPIPS. These metrics quantify appearance alignment in pixel space or feature space and are sensitive to non-essential factors such as cropping, global scaling, and label placement, which are employed in text-to-image generation tasks (Huang et al., 2023; Li et al., 2025a; Wei et al., 2026). We therefore use them as diagnostic signals rather than primary measures of geometric correctness.

## B. Comparison with Existing Benchmarks

Table 8 positions **Faire-Bench** among representative multimodal math benchmarks along two dimensions that matter for constructive geometry reasoning. First, *capability coverage* asks whether a benchmark requires (i) solving the math problem, (ii) producing a diagram as an output, and (iii) solving in an interleaved text–image manner. Second, *supervision modality* asks whether instances provide aligned *text*, *image*, and *code* signals. We report each attribute as a binary indicator for clarity.

**From answer selection to constructive evidence.** Most multimodal math benchmarks focus on problem solving with static visual inputs and evaluate models primarily by answer correctness. Even when generation is considered, the output is rarely required to be a *constructive artifact* that can be executed and checked against geometric constraints. Faire-Bench instead evaluates a model's ability to act: it must produce both a correct solution and an executable construction that reconstructs the intended geometric state.

**Tri-modal supervision for verifiability.** A central limitation in prior benchmarks is the lack of aligned program supervision. Without code, it is difficult to verify whether the reasoning and the final diagram are mutually consistent, and failures often collapse into unscored hallucinations. Faire-Bench provides aligned *text*, *image*, and *code* for every instance, enabling deterministic executability checks and geometry-aware verification beyond surface similarity.

**Interleaved solving with executable grounding.** Benchmarks that include drawing typically emphasize visualization quality, but do not couple drawing with the full problem-solving process, and thus do not support evaluating interleaved reasoning and construction as a single coherent behavior. Faire-Bench explicitly supports interleaved text–image solving and grounds that behavior in executable code, which makes intermediate constructions auditable rather than merely narrated.

**Takeaway.** Together, these axes make Faire-Bench a constructive and verifiable benchmark that unifies reasoning, drawing, and interleaved solving under tri-modal supervision, complementing understanding-oriented math datasets and filling the gap between visual reasoning and executable geometry.

---

**VLM-as-Verifier Prompt for Diagram Fidelity**

You are a GeoGebra diagram quality auditor with strong mathematical expertise. Your task is to verify whether a rendered diagram is *mathematically correct*, *logically consistent*, and *visually usable*, by jointly examining the problem image, the solution text (including reasoning flow), and the rendered diagram produced from GeoGebra code.

**Inputs**
You are given: (1) the original problem image as reference; (2) the solution text, including step-by-step reasoning and embedded `geogebra` code blocks; (3) the rendered diagram generated from the code (**this is the primary object to be evaluated**).

**Verification Checklist**
Perform the following checks and decide whether the rendered diagram is usable.
**(1) Contextual intent alignment** First identify the drawing intent from the surrounding solution text. If the text explicitly limits scope (*e.g.,* "we illustrate case 1 only" or "we draw the translated figure"), do *not* penalize missing elements outside this declared scope. The diagram is considered aligned as long as it faithfully reflects what the current step claims to construct.
**(2) Mathematical correctness and geometric completeness** Verify that all required points, lines, curves, or function plots needed for the reasoning step are present. Check that defined mathematical relations (*e.g.,* incidence, collinearity, relative position, shape properties) are correctly instantiated. The geometric relations expressed by the diagram must be consistent with the solution text and the problem specification.
**(3) Visual connectivity and structural integrity** Inspect all auxiliary constructions such as dashed lines, perpendiculars, or projection guides. Auxiliary elements must be physically connected to their target objects (axes, curves, points). If any auxiliary line is visibly disconnected or floating, mark the diagram as invalid.
**(4) Visual clarity and readability** Assess whether the diagram is clear and interpretable. Penalize severe clutter, overlapping elements, blurred rendering, or occluded labels. Key points and annotations should be legible and unambiguous.
**(5) Allowed visual deviations** Do not penalize distortions caused by automatic axis scaling in GeoGebra. As long as topological relations are preserved (*e.g.,* an ellipse remains a closed curve), differences in apparent proportions are acceptable.

**Output Format**
Return a binary decision: `1` if the diagram is usable; `0` otherwise.
If returning `0`, provide a brief justification in the form: `[Error Type] Specific description`,
for example: `[Mathematical Error] Point (3,2) is shown on line` $y = x - 2$ `but does not satisfy the equation`, or `[Visual Error] The dashed projection from point A does not connect to the x-axis`.

---

*Figure 9.* Prompt used for the *VLM-as-Verifier* in Faire. The verifier performs pixel-level, mathematical, and logical consistency checks to determine whether a rendered diagram faithfully supports the intended reasoning step.

## C. Verifier Prompt Specifications

### C.1. VLM-as-Verifier.

Figure 9 shows the prompt used for the vision-language verifier in Faire. Unlike generic preference-based judges, this verifier is explicitly instructed to assess diagram usability through a structured checklist that combines contextual intent, mathematical validity, geometric connectivity, and visual readability. Crucially, the verifier is designed to tolerate benign visual distortions introduced by automatic coordinate scaling, while remaining strict about topological correctness and logical consistency. This design enables pixel-level yet mathematically grounded supervision, forming a key component of the post-generation verification pipeline.

### C.2. Parser-as-Verifier.

Figure 10 presents the prompt used for the parser-based verifier in Faire. This verifier targets semantic consistency between the symbolic geometric description and the reasoning process, operating at a level orthogonal to pixel-based inspection.

Unlike the VLM-based verifier, which evaluates rendered diagrams, the parser-based verifier audits a structured textual description of the construction (`ggb_parser`) that enumerates geometric entities, relations, and attributes. The prompt enforces a fine-grained, checklist-driven comparison between the parsed geometric specification and the mathematical reasoning steps, covering element existence, relational correctness, hidden-structure integrity, and internal consistency.

By grounding verification in explicit coordinates, equations, and relational constraints, the parser-based verifier detects failures that are visually plausible yet semantically incorrect. It therefore instantiates the semantic alignment signal in our tri-perspective verification framework, complementing visual usability checks and formal geometric assertions.

You are an expert verifier specializing in analytic geometry and step-by-step mathematical reasoning. Your role is to act as a **semantic geometry validator** that audits whether a constructed diagram faithfully instantiates the reasoning process described in the solution.

**Task Description**

You are given: (1) a problem image; (2) a detailed solution text containing one or more `ggb_parser` blocks.

Each `ggb_parser` block is a *textual description* of a geometric construction, listing points, lines, curves, intersections, attributes (coordinates, domains, visibility), and relations. It is **not executable code**. Your task is to verify whether this description is semantically consistent with both the problem statement and the mathematical reasoning steps.

**Verification Checklist**

For *each* `ggb_parser` block, perform the following checks.

*(I) Element Accuracy*

**Points.** Verify that all point coordinates match those derived in the solution. List where each point is expected to lie (on which curve, line, or segment), then check whether the parser description is consistent. Missing or misplaced points constitute an error.

**Curves.** Check whether curve definitions (equations, branches, domains) exactly match the derivation. For example, if the solution specifies a restricted branch of a conic, the parser must reflect the same restriction.

**Lines.** Verify that line equations (slope, intercept, tangency) match the computed results.

**Hidden Elements.** For invisible objects, ensure that required connections are still materialized. Hidden carrier lines must still induce the necessary visible segments to avoid disconnected points.

**Angles.** If angle elements appear in the parser, verify that they are explicitly required by the problem or solution. Any unnecessary non-right angle is considered incorrect.

**Length Relations.** Exact numeric lengths need not match, but relative ordering must. For example, if $AC > AB > BC$ in the solution, the construction must preserve this inequality.

**Polygons.** Verify polygon naming consistency using coordinate-based cyclic ordering. Determine the true geometric vertex cycle via centroid and polar-angle sorting, and check whether the given name matches up to rotation or reversal. Incorrect adjacency implies an invalid polygon.

*(II) Relational Consistency*

Check that all incidence relations (point-on-line, point-on-curve) and proven geometric relations (collinearity, tangency, concurrency) are explicitly reflected in the parser description.

*(III) Internal Consistency*

Extract the diagram description sentence (*e.g.,* "the figure below shows..."). Verify that every described element appears in the parser and that no contradictions exist.

*(IV) Completeness*

Based on the original problem image and statement, list all required geometric elements. If any required element is missing across all parser blocks, the construction is invalid.

**Output Format**

For each parser block, produce a structured *Diagram Verification Report* including: (1) element-level analysis; (2) relation-level consistency; (3) internal consistency; (4) completeness; (5) a list of detected issues (or "no issues found").

Conclude with three binary judgments: (a) corresponds to the solution process; (b) satisfies the problem intent; (c) is geometrically correct.

**Final Decision**: output `1` if and only if all three judgments are positive; otherwise output `0`. If multiple parser blocks exist, the final score is `1` only if all blocks pass.

*Figure 10.* Prompt used for parser-based semantic verification, which evaluates whether a textual geometric description faithfully instantiates the intended reasoning and relations.

### C.3. GGB-as-Verifier.

Figure 11 shows the prompt used for the GeoGebra-based verifier in Faire. This verifier provides a deterministic correctness signal by checking whether the generated construction satisfies formal geometric properties implied by the diagram.

Unlike the visual and parser-based verifiers, which rely on perceptual or semantic judgments, the GGB-as-Verifier operates by synthesizing explicit Boolean assertions from the observed geometric relations. Given the rendered diagram and the corresponding GeoGebra commands, the verifier translates visual relations (*e.g.,* collinearity, perpendicularity, tangency, or region membership) into executable GeoGebra predicates.

These predicates are then evaluated directly by the GeoGebra kernel, yielding a binary outcome that reflects mathematical truth rather than appearance. As such, the GGB-as-Verifier instantiates the *formal validity* signal in our tri-perspective verification framework, preventing self-consistent but geometrically incorrect constructions from receiving reward.

---

**GGB-as-Verifier Prompt for Formal Geometric Validity**

You are an expert in GeoGebra and formal geometric verification. Your task is to analyze a rendered geometric diagram together with its corresponding GeoGebra construction, and to generate executable verification commands that test whether the construction satisfies the geometric relations implied by the diagram.

**Inputs**

You are given: (1) an image of the rendered geometric construction; (2) a set of GeoGebra commands (`user_construction`) that produced the diagram.

**Task Description**

Your goal is to identify key geometric properties visible in the diagram and translate them into formal GeoGebra boolean expressions. These expressions will be executed by the GeoGebra kernel to verify whether the construction is mathematically correct.

**Verification Procedure**

**(1) Visual relation analysis** Inspect the diagram and identify essential geometric relations, such as: points lying on lines or curves, intersections, collinearity, concurrency, parallelism, perpendicularity, tangency, concyclicity, or region inclusion.

**(2) Assertion synthesis** For each identified relation, generate a corresponding GeoGebra boolean predicate that evaluates to `true` if the relation holds. Each predicate must be directly executable on the provided construction.

**(3) Allowed primitives** You may use the following GeoGebra verification functions:

- `AreCollinear(<Point>, <Point>, <Point>)`

- `AreConcurrent(<Line>, <Line>, <Line>)`

- `AreConcyclic(<Point>, <Point>, <Point>, <Point>)`

- `AreParallel(<Line>, <Line>)`

- `ArePerpendicular(<Line>, <Line>)`

- `IsTangent(<Line>, <Conic>)`

- `IsInRegion(<Point>, <Region>)`

**Output Format**

Return a JSON object with a single field:

- `"verification_code"`: a list of GeoGebra boolean expressions, where each expression verifies one geometric property.

All generated expressions must be valid GeoGebra commands and should collectively cover the essential geometric relations depicted in the diagram.

---

*Figure 11.* Prompt used for the *GGB-as-Verifier* in Faire. The verifier synthesizes executable GeoGebra assertions from visual relations and evaluates them deterministically to enforce formal geometric correctness.

## D. Where Functional Alignment Matters Most

Figure 12 reveals a clear visual dichotomy between *systematic failure* and *robust grounding*. Across geometry-heavy categories, several strong generalist MLLMs form large pale regions in the heatmap, indicating not sporadic errors but a structural inability to ground deductions in constructed geometric states. For instance, on *Lines & Coordinates*, Gemini-2.5 (Comanici et al., 2025) reaches only 12.37 and GPT-4o (Hurst et al., 2024) drops to 7.56, whereas Faire achieves 59.11, representing nearly a five-fold improvement. A similar pattern appears on *Conics: Circle & Ellipse*, where Gemini-2.5 (Comanici et al., 2025) scores 11.49 while Faire exceeds it by more than three times. Crucially, these collapses align with categories where deductions are tightly state-dependent: the next reasoning step is valid only if the constructed configuration is exact. The pale regions therefore diagnose a failure of distributional alignment: models reproduce the appearance of interleaved diagrams, yet the instantiated relations cannot reliably support subsequent reasoning.

In contrast, Faire forms a uniformly dark column across all categories, without localized degradation. It maintains high scores on *Functions & Graphs* (70.93), *Exponential/Log/Power Functions* (59.52), and *Lines & Coordinates* (59.11), demonstrating category-robust reasoning grounded in executable geometric states. This consistency reflects functional alignment: construction is not a visual byproduct, but a reliable intermediate state actively used by the solver. These results establish a methodological conclusion. When interleaving is optimized only at the distributional level, increasing model scale does not prevent geometric collapse. By contrast, enforcing functional alignment through post-generation verification enables a smaller model to systematically outperform much larger generalist MLLMs. This shift, from imitating interleaving to internalizing construction as reasoning, defines the core advantage of Faire.

| | Gemma-3 | GPT-4o | InternVL | QWen-2.5-VL | KimiVL | GLM-4.1V | QWen-3-VL | Gemini-2.5 | GPT-5.1 | GPT-5.2 | Faire |
|---|---|---|---|---|---|---|---|---|---|---|---|
| Lines & Coordinates | 4.12 | 7.56 | 7.22 | 6.19 | 11.68 | 10.65 | 22.68 | 12.37 | 25.09 | 26.80 | 59.11 |
| Conics: Circle & Ellipse | 6.90 | 2.30 | 4.60 | 9.20 | 11.49 | 8.05 | 10.34 | 11.49 | 11.49 | 20.69 | 35.63 |
| Conics: Hyperbola & Parabola | 2.34 | 2.34 | 7.60 | 10.53 | 11.11 | 11.70 | 14.62 | 17.54 | 16.37 | 12.28 | 38.60 |
| Lines: Angles & Parallels | 3.56 | 2.22 | 5.78 | 4.44 | 9.33 | 18.67 | 7.11 | 8.44 | 8.00 | 14.67 | 29.33 |
| Circle Geometry | 1.28 | 3.85 | 10.26 | 8.33 | 7.69 | 12.18 | 7.05 | 25.00 | 15.38 | 24.36 | 54.49 |
| Quadrilaterals & Polygons | 2.16 | 6.06 | 6.28 | 6.49 | 10.39 | 11.47 | 6.93 | 13.85 | 19.26 | 21.21 | 49.57 |
| Triangles: Congruence & Similarity | 1.90 | 3.57 | 8.93 | 7.14 | 7.62 | 11.07 | 10.95 | 14.64 | 12.02 | 18.93 | 47.86 |
| Exp/Log/Power Functions | 4.76 | 4.76 | 11.90 | 11.90 | 7.14 | 21.43 | 30.95 | 26.19 | 35.71 | 33.33 | 59.52 |
| Functions & Graphs | 4.13 | 9.87 | 14.40 | 8.40 | 12.13 | 12.53 | 21.07 | 19.60 | 26.27 | 27.87 | 70.93 |
| Triangle Trigonometry | 3.92 | 3.92 | 0.00 | 7.84 | 7.84 | 17.65 | 15.69 | 3.92 | 5.88 | 7.84 | 47.06 |

*Figure 12.* Category-wise verification scores.

## E. More Aha Moment Cases: From Format Imitation to Functional Alignment

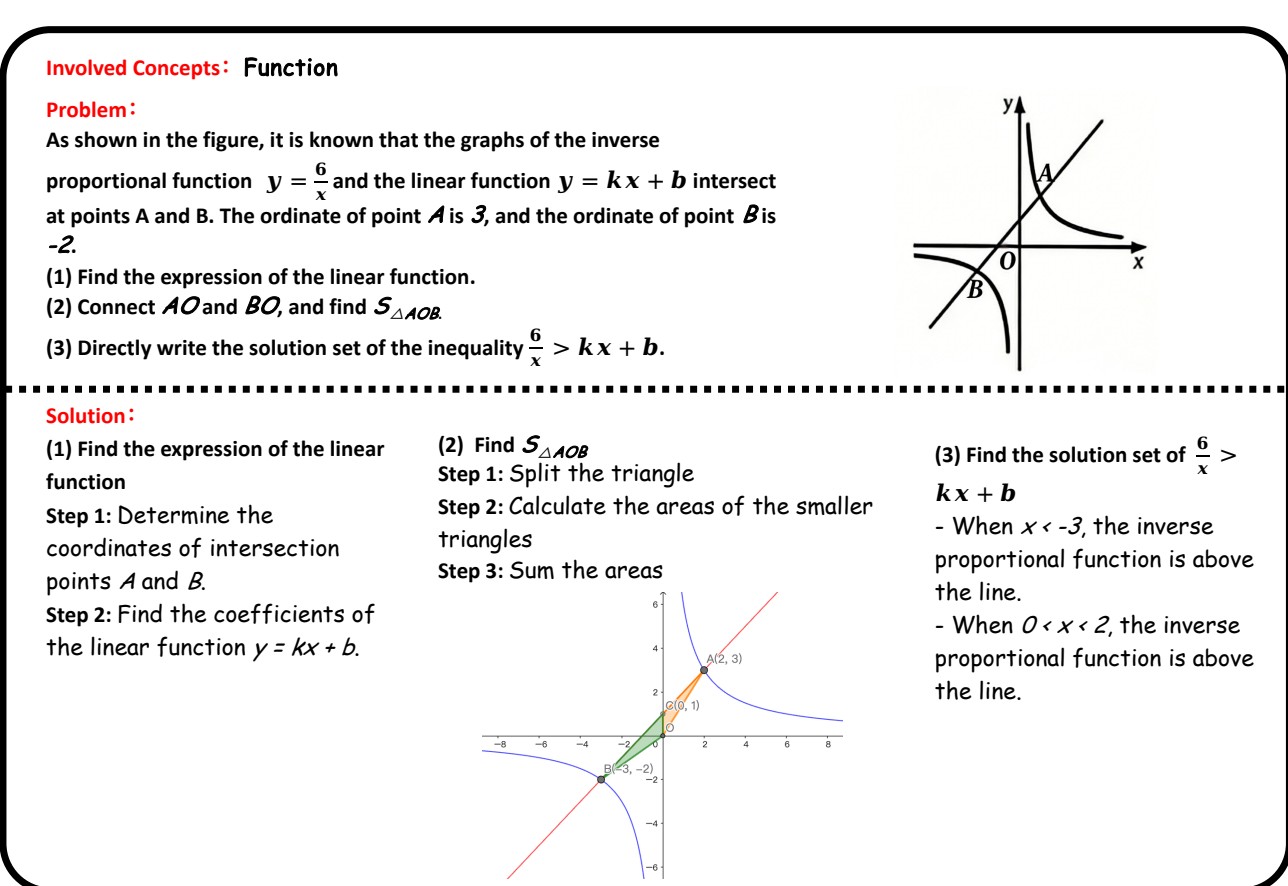

**Involved Concepts:** Function

**Problem:**
As shown in the figure, it is known that the graphs of the inverse proportional function $y = \frac{6}{x}$ and the linear function $y = kx + b$ intersect at points A and B. The ordinate of point *A* is *3*, and the ordinate of point *B* is *-2*.
(1) Find the expression of the linear function.
(2) Connect *AO* and *BO*, and find $S_{\triangle AOB}$.
(3) Directly write the solution set of the inequality $\frac{6}{x} > kx + b$.

**Solution:**
(1) Find the expression of the linear function
Step 1: Determine the coordinates of intersection points *A* and *B*.
Step 2: Find the coefficients of the linear function $y = kx + b$.

(2) Find $S_{\triangle AOB}$
Step 1: Split the triangle
Step 2: Calculate the areas of the smaller triangles
Step 3: Sum the areas

(3) Find the solution set of $\frac{6}{x} > kx + b$
- When $x < -3$, the inverse proportional function is above the line.
- When $0 < x < 2$, the inverse proportional function is above the line.

*Figure 13.* **Interleaved function visualization (line + reciprocal curve).** Faire constructs both graphs in the same coordinate system and marks the key points needed by the reasoning, so the option decision is supported by a checkable geometric state.

We analyze five representative cases (Figures 13–17) to illustrate how interleaved reasoning succeeds or fails in practice. Across these examples, Faire consistently treats the diagram as an executable intermediate state, using construction feedback to ground and constrain subsequent deductions. Meanwhile, when compared with strong general-purpose models (*e.g.,*

**Involved Concepts:** Analytic Geometry

**Problem:**

As shown in the figure, the vertex *A(−2,4)* of the right triangle *OAB* lies on the parabola $y = ax^2$. Rotate *Rt△OAB* *90°* clockwise around point *O* to obtain *Rt△OCD*. The side *CD* intersects the parabola at point *P*. Find the coordinates of point *P*.

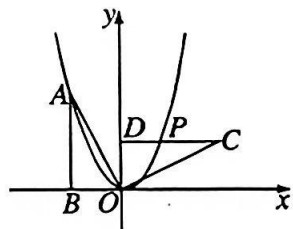

**Solution:**

**Step 1: Find the equation of the parabola**
Given point *A(−2,4)* lies on the parabola *y=ax²*, substitute into the equation: *4 = a×(−2)² ⟹ 4 = 4a ⟹ a = 1*
Thus, the equation of the parabola is *y=ax²*.

**Step 2: Find the coordinates of points *C* and *D* after rotation**
First, determine the coordinates of point *B*: Since *A(−2,4)* and *AB⊥x-axis*, *B(−2,0)*.
The coordinate transformation for a 90° clockwise rotation around the origin is *(x,y) → (y,−x)*:
Point *A(−2,4)* rotates to *C(4,2)*;
Point *D(0,2)* rotates to *D(0,2)*.

**Step 3: Find the equation of line *CD***
Points *C(4,2)* and *D(0,2)* have the same y-coordinate, so line *CD* is horizontal, with equation *y=2*.

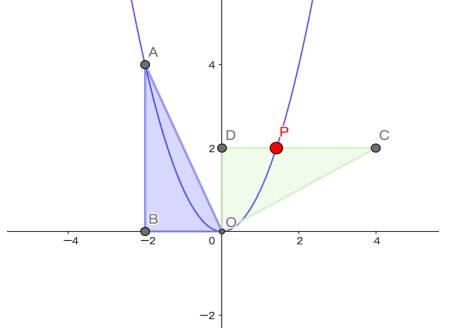

**Step 4: Find the coordinates of intersection point *P***
Solve the system of equations for the parabola and line *CD*: *y=x²; y=2 ⟹ x=2 or x=−2.*
From the figure, *P* is in the first quadrant, so take *x=2*. Thus, the coordinates of *P* are *(2,2)*.

*Figure 14.* **Analytic-geometry construction with projections.** The diagram materializes auxiliary points and projection relations, turning abstract constraints into explicit incidences that the subsequent deduction can rely on.

Gemini and GPT), failures typically arise from weaker text–diagram coupling: diagrams are produced as illustrative artifacts rather than as binding states that actively guide the reasoning process. In contrast, Faire consistently treats the diagram as an executable intermediate state, using it to support and constrain subsequent reasoning.

In problems where correct solutions hinge on precise geometric instantiation, such as function intersections, projections, and auxiliary constructions (Figures 13, 14, and 15), Faire produces diagrams whose geometric relations directly justify the next step in the solution. Intersections, alignments, and auxiliary points are not merely visual cues but function as verifiable evidence. By comparison, baseline models often generate visually plausible diagrams that fail to encode the relations their own reasoning implicitly assumes, causing downstream deductions to rest on unsupported geometric states.

A more revealing failure mode appears when symbolic answers are correct, but constructions are not. Figure 16 shows that Gemini-2.5 reaches the correct logical conclusions, yet its generated diagram violates orientation and connectivity constraints, rendering it unusable as a proof artifact. GPT-5 fails more fundamentally, producing both incorrect reasoning and invalid constructions. These cases demonstrate that the correctness of the final answer does not imply the correctness of the constructed geometric state. Only Faire satisfies symbolic reasoning, geometric construction, and visual consistency simultaneously.

Equal-length relations are mistaken for midpoint relations, auxiliary lines are introduced without the constraints they are meant to encode, and errors propagate across steps. Here, interleaving becomes a liability rather than an aid, confirming that SFT optimizes surface alternation instead of functional dependency.

Faire resolves these failures by enforcing post-generation verification at each step. Every constructed diagram is checked against the intended geometric relations before it is used for further reasoning. As a result, incorrect constructions are rejected early, preventing silent error accumulation. The diagram is no longer a decorative output but a proof-carrying state that constrains the reasoning process.

**Involved Concepts**： **Plane Geometry**

**Problem**：

As shown in the figure, in $\triangle ABC$, $AD$ is the median. Point $E$ lies on $AB$, and $AD$ intersects $CE$ at point $F$. Given $AE=EF$, prove that $AB=CF$.

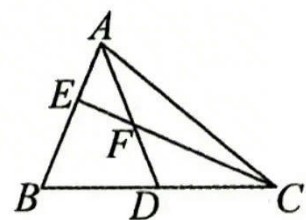

**Solution**：

**Step 1: Construct the auxiliary line and prove triangle congruence**
Extend $AD$ to point $G$ such that $DG=AD$, then connect $BG$ and $CG$.

- Since $AD$ is the median of $\triangle ABC$, $BD=CD$.
- In $\triangle ADB$ and $\triangle GDC$: $AD=DG$ (by construction), $\angle ADB=\angle GDC$ (vertical angles are equal), $BD=CD$ (definition of a median). By the SAS congruence criterion, $\triangle ADB \cong \triangle GDC$. Thus, $AB=GC$ and $\angle BAD=\angle G$.

**Step 2: Use angle relationships to prove an isosceles triangle**
Given $AE=EF$, we have $\angle AFE=\angle FAE$.
$\angle AFE$ and $\angle GFC$ are vertical angles, so $\angle AFE=\angle GFC$, hence $\angle GFC=\angle FAE$.
Since $\angle FAE=\angle BAD=\angle G$, it follows that $\angle GFC=\angle G$. Thus, $\triangle GFC$ is isosceles, so $GC=CF$.

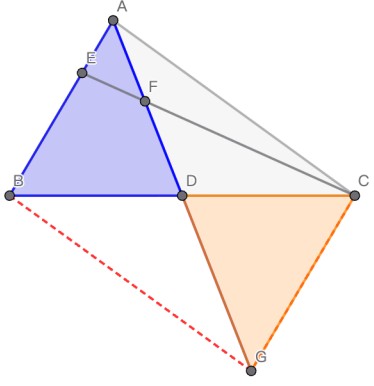

**Step 3: Conclusion**
From $AB=GC$ and $GC=CF$, by substitution, we get $AB=CF$.

*Figure 15.* **Plane-geometry construction via auxiliary intersections.** The construction exposes hidden structure (intersections and induced sub-triangles), providing a concrete state for validating proportional or area arguments.

Finally, Figure 17 shows that functional grounding remains critical even when algebraic manipulation is correct. Both GPT-5 and Gemini-2.5 derive valid intersection conditions, yet select the wrong graph due to a failure to reason globally about quadrant structure. By explicitly constructing and inspecting the geometric configuration, Faire arrives at the correct choice. This highlights a broader point: interleaved reasoning succeeds not by adding diagrams, but by making constructions functionally indispensable to deduction.

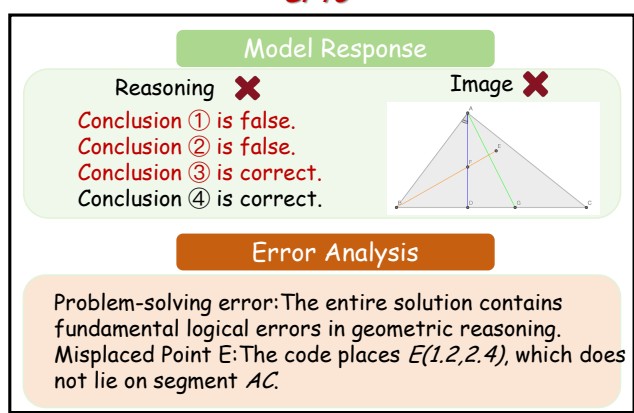

### Problem Description

As shown in the figure, in △ABC, ∠BAC=90°, AD⊥BC. The angle bisector BE of ∠ABC intersects AD at point F, and AG bisects ∠DAC. The following conclusions are given:
① ∠BAD=∠C; ② ∠AEF=∠AFE; ③ ∠EBC=∠C; ④ AG⊥EF.

### Our Model

**Model Response**

Reasoning ✔          Image ✔

Conclusion ① is correct.
Conclusion ② is correct.
Conclusion ③ is false.
Conclusion ④ is correct.

### Gemini2.5

**Model Response**

Reasoning ✔          Image ✖

Conclusion ① is correct.
Conclusion ② is correct.
Conclusion ③ is false.
Conclusion ④ is correct.

**Error Analysis**

Incorrect Orientation: The coordinate system rotates the original triangle, breaking alignment with the problem's diagram.
Cluttered Visuals: Non-essential auxiliary lines are not hidden, creating visual noise.

### GPT5

**Model Response**

Reasoning ✖          Image ✖

Conclusion ① is false.
Conclusion ② is false.
Conclusion ③ is correct.
Conclusion ④ is correct.

**Error Analysis**

Problem-solving error: The entire solution contains fundamental logical errors in geometric reasoning.
Misplaced Point E: The code places E(1.2,2.4), which does not lie on segment AC.

*Figure 16.* **Reasoning–diagram consistency comparison on a judgement problem.** Faire keeps deductions and construction aligned to the same geometric state; baselines may produce answer-correct text with diagram drift (orientation/clutter) or fail in both reasoning and construction.

## Question

In the same plane rectangular coordinate system, the approximate graphs of the functions $y = -x + k$ and $y = \frac{k}{x}$ (where $k$ is a constant and $k \neq 0$) are

( )

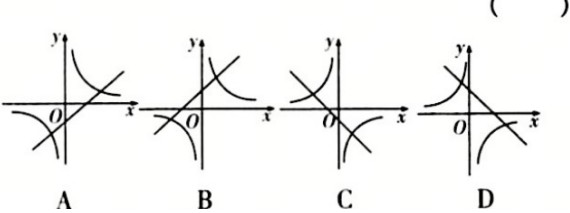

A      B      C      D

## OURS

This question examines the properties of the graphs of a linear function and an inverse proportional function. It requires analyzing the characteristics of both function graphs based on the sign of the parameter $k$ and matching them to the correct option.

**Step 1:** Analyze the linear function $y = -x + k$

The slope of the linear function is $-1$, so the line decreases from left to right and must pass through the second and fourth quadrants. The intercept is $k$, which determines the intersection with the y-axis……

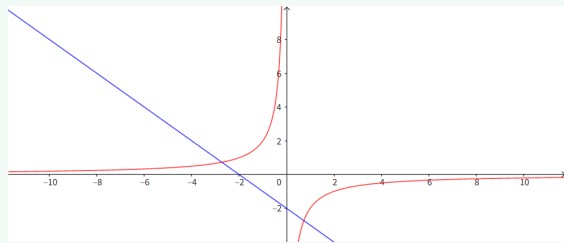

## Conclusion

Only Option C matches the graph characteristics of both functions when k < 0.
*The correct answer is C.*

## GPT-5

Sketch the approximate positions of the line $y = -x + k$ and the hyperbola $y = \frac{k}{x}$ (with $k < 0$) in the same coordinate system, and select the correct graph.

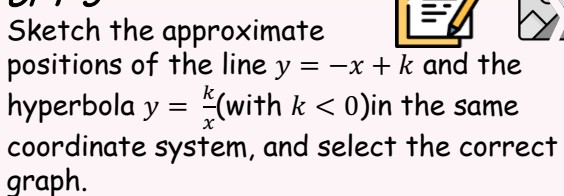

Find the intersection points by equating the two expressions:
$$-x + k = \frac{k}{x}$$
$$x^2 - kx + k = 0$$
The discriminant is:
$$\Delta = k^2 - 4k = k(k - 4)$$
For two intersection points, we need $\Delta > 0$, which implies $k > 4$ or $k < 0$.
……
……and third quadrants……Among the four options, *the one that fits is D*.

## Gemini-2.5

Here is a step-by-step analysis to determine the correct graph:
The problem asks to find the correct graphical representation for the two functions y = -x + k and y = k/x on the same coordinate plane, where k is a non-zero constant.
Let's analyze the properties of these two functions based on the sign of the constant k.
**Case 1:** $k > 0$
**Linear Function** $(y = -x + k)$:
**Slope:** The slope is -1, which is negative. This means the line must be decreasing (it goes down from left to right).
……
**Conclusion:**
The only case that results in a consistent graph among the given options is when k > 0. This corresponds to Option D.
Therefore, *the correct answer is D*

*Figure 17.* **Function graph selection: grounded structure versus ungrounded calculation.** Faire anchors the decision in a state-faithful plot under the given sign constraints, while large generalist models may compute extensively yet still select an option inconsistent with the intended configuration.

