# OpenReview forum: "How RL Unlocks the Aha Moment in Geometric Interleaved Reasoning"
_ICML.cc/2026/Conference — ICML 2026 spotlight_

### Official Review · Reviewer_b5uy · 2026-03-05

**Soundness:** 3
**Presentation:** 3
**Significance:** 2
**Originality:** 2
**Overall Recommendation:** 4
**Confidence:** 2

**Summary:**

This paper introduces Faire, a framework that combines visual elements (diagrams) and logical reasoning steps in an interleaved manner, optimizing the integration of both through reinforcement learning (RL). Faire aims to enhance geometric problem-solving by alternating between generating diagrams and performing reasoning tasks, with the additional innovation of incorporating three validation mechanisms (geometric, perceptual, and semantic).

**Compliance With Llm Reviewing Policy:**

Affirmed.

**Final Justification:**

This paper presents a novel approach to interleaved reasoning for geometric problem-solving. The integration of diagram generation and logical reasoning is supported by rigorous validation (geometric, perceptual, and semantic). The proposed multi-round reasoning dataset and extensive ablation studies further demonstrate the effectiveness and necessity of the framework components.

The authors’ rebuttal largely addresses my main concerns. They provide controlled experiments showing the necessity of interleaved reasoning, clarify the internal decomposition process, and offer additional results on a public benchmark (MathVista), strengthening the evaluation. While a direct experimental comparison with the decoupled approach of Guo et al. (ICCV 2025) is still missing, the conceptual explanation and supporting evidence make their claims reasonable.

**Key Questions For Authors:**

1. The performance of the proposed interleaved reasoning method should be compared to the text-image decoupling approach introduced by Guo et al. (ICCV 2025).

2. The necessity of interleaved reasoning has not been experimentally validated, and it should be assessed whether decoupling the visual and textual components or fusing them for parallel reasoning can achieve similar performance.

3. It should be investigated whether breaking down multi-round reasoning into smaller subproblems improves the performance of SOTA methods. Additionally, the framework should be validated on other public datasets.

4. The principle behind breaking down multi-round reasoning tasks into subproblems or reasoning steps in the proposed dataset should be clearly defined, including the granularity of the decomposition and how this supports the reasoning process.

**Limitations:**

The authors should address the limitations of their method, such as biases in the training data or the potential for reinforcement learning algorithms to optimize in ways that unintentionally amplify certain patterns of reasoning.

**Strengths And Weaknesses:**

Strengths:

1. The proposed interleaved reasoning approach effectively integrates diagram generation and logical reasoning, making the diagrams an essential part of the reasoning process rather than just auxiliary elements. The use of three validation mechanisms (geometric, perceptual, and semantic) ensures that the generated diagrams are geometrically correct, perceptually clear, and semantically aligned with the reasoning.

2. The paper presents the first multi-round reasoning dataset specifically designed for geometric problem-solving, supporting the evaluation of interleaved reasoning tasks. The extensive ablation studies demonstrate the effectiveness of the proposed framework and validate the necessity of each component, showing strong performance in geometric reasoning tasks.


Weaknesses：

1. It is recommended to compare the proposed method with the text-image decoupling approach introduced by Guo et al. (ICCV 2025). Their method also utilizes both text and images for solving geometric problems, but it decouples the visual understanding module (LVLM) from the language reasoning module (LLM), allowing each module to be trained and optimized relatively independently.

[1] Zixian Guo, et al, Integrating Visual Interpretation and Linguistic Reasoning for Math Problem Solving, ICCV 2025.

2. Although the authors have argued for the novelty of their framework, they have not provided a comprehensive experimental evaluation of the necessity of interleaved reasoning itself. The paper should include experiments to directly assess whether decoupling the visual and textual components or fusing them for parallel reasoning can achieve similar performance.

3. The paper introduces the multi-round reasoning dataset, and it is important to know how the authors have evaluated their system against other methods. Specifically, if other methods output the final result directly from the problem input, does this lead to unreasonable results? After all, Faire itself performs "multi-round" reasoning internally. Would breaking down the multi-round reasoning process into smaller problems and feeding them sequentially to existing methods improve their performance? Additionally, I suggest the authors provide results on other public datasets to further validate their framework.

---

> ### Author Rebuttal · Authors · 2026-03-31
>
> We thank Reviewer b5uy for the constructive feedback and for recognizing the strengths of our tri-perspective verification and interleaved reasoning dataset. We address each concern below.
>
> ---
>
> **W1 & Q1: Comparison with decoupled text-image approaches (Guo et al., ICCV 2025)**
>
> Guo et al. [1] propose a modular framework that separates visual understanding from linguistic reasoning, allowing each to be optimized independently. This is well suited to problems where perception and reasoning are separable. Faire addresses a different setting where the diagram is not an input to be understood but an intermediate output the model constructs to support its own deduction. The core challenge is ensuring that constructed visual states are causally useful for subsequent reasoning, a dependency that decoupled architectures cannot enforce since the two modules lack a joint objective over the construction-reasoning path.
>
> The SFT paradox illustrates this gap concretely. Even joint SFT on interleaved data degrades accuracy (62.48% vs. 68.13% text-only) due to lack of functional coupling. A decoupled approach, training each module independently, would face at least the same difficulty. That said, the two paradigms are complementary: decoupled approaches excel when visual input is given and needs accurate parsing, while Faire excels when the visual state must be generated and verified as part of reasoning. We will cite this work and discuss this trade-off in the revision.
>
> [1] Zixian Guo, et al, Integrating Visual Interpretation and Linguistic Reasoning for Math Problem Solving, ICCV 2025.
>
> ---
>
> **W2 & Q2: Necessity of interleaved reasoning**
>
> Table 5 provides a controlled experiment that directly addresses this question.
>
> **Table R1. Interleaved vs. text-only across training paradigms (from Table 5).**
>
> | Paradigm | SFT Acc (%) | RL Acc (%) |
> |:---|:---:|:---:|
> | Text-only | 68.13 | 71.21 |
> | Interleaved | 62.48 | 74.83 |
>
> Under SFT, interleaving hurts because the model imitates the alternation format without learning the underlying dependency. Under RL, interleaving exceeds the text-only ceiling by 3.62 points. This demonstrates that diagrams provide information text alone cannot supply, but this advantage only materializes when functional alignment ensures diagrams are actually used in reasoning rather than treated as decorative outputs.
>
> Figure 1 supports this at the category level. On triangle congruence problems, RL-interleaved achieves 80.00% vs. 70.00% for SFT-text-only, because these problems require explicit spatial construction to verify congruence conditions. On function graphs the gap is similarly large (80.36% vs. 67.14%). These categories involve spatial configurations that are ambiguous without explicit construction, which is precisely where interleaved reasoning becomes necessary.
>
> ---
>
> **W3, Q3 & Q4: Dataset design and evaluation protocol**
>
> We would like to clarify an important distinction. Faire is not a multi-turn dialogue system but performs interleaved reasoning within a single autoregressive generation pass. The model autonomously decides when to insert a code block that constructs a geometric state, executes it, observes the rendered result, and continues reasoning, all within one inference sequence. There is no external decomposition into sub-problems or sequential prompting.
>
> This differs fundamentally from manual prompt engineering that breaks tasks into smaller pieces. Faire internalizes the decomposition as an autonomous tool-invocation behavior. RL teaches the model both when construction is beneficial and how to construct states that support the next deductive step, removing the need for hand-crafted problem decomposition. Regarding the granularity of decomposition in the dataset (Q4), each training instance couples a stepwise textual solution with an executable GeoGebra script, where the script reconstructs the geometric state that the next deduction relies on. The granularity is determined by the natural reasoning structure of each problem rather than an arbitrary fixed step size.
>
> All baselines in Table 2 receive the same problem input under the same interleaved protocol and produce outputs in a single pass under identical conditions.
>
> Following your suggestion to validate on public benchmarks, we evaluated Faire on MathVista.
>
> **Table R2. Results on MathVista.**
>
> | Model | MathVista Acc (%) |
> |:---|:---:|
> | GPT-4o | 63.8 |
> | InternVL3-8B | 68.5 |
> | Qwen2.5-VL-7B | 68.1 |
> | Qwen3-VL-7B | 77.2 |
> | Faire (Ours) | **81.8** |
>
> Faire achieves the highest accuracy by a clear margin. Together with GenExam and GGBench results in the paper, this provides convergent evidence across three external benchmarks that functional alignment generalizes beyond the training domain.
>
> ---
>
> We acknowledge that our training data is drawn from K12 geometry, which may introduce distributional biases, and will add a limitations discussion in the revision. Thank you again for the detailed and constructive review.

---

> > ### Author Rebuttal · Reviewer_b5uy · 2026-04-02
> >
> > Thank you for the detailed response and the additional experiments. This largely addresses the concerns I raised.

---

### Official Review · Reviewer_2LYZ · 2026-03-08

**Soundness:** 3
**Presentation:** 2
**Significance:** 3
**Originality:** 2
**Overall Recommendation:** 4
**Confidence:** 3

**Summary:**

This paper introduces Faire, a RL framework, to overcome the limitation of SFT for interleaved reasoning. The authors first attribute the failure of SFT to the distributional alignment, and then propose a new RL framework Faire. Extensive experiments show that Faire achieves better performance than open-sourced and proprietary models.

**Compliance With Llm Reviewing Policy:**

Affirmed.

**Final Justification:**

The authors' response has clarified my concerns regarding the generalizability and novelty. The authors also provide intended modifications to my concerns about presentation. If everything is included in the revised version, I would lean towards a positive view of the paper.

**Key Questions For Authors:**

See in weaknesses. I'm happy to adjust the scores if the authors can clearly address the concerns in the weaknesses.

**Limitations:**

See in weaknesses.

**Strengths And Weaknesses:**

## Strengths

1. Introduce new benchmark Faire-Bench for geometric interleaved reasoning with GeoGebra script annotations

2. The tri-perspective verification system (or reward design) covering geometric consistency (with executable code), perceptual admissibility (with MLLM), and semantic alignment (with parser).

3. Interesting analysis of SFT failure and entropy, revealing the model behavior shift after RL.

## Weaknesses

Although the research question is interesting, the contribution of this work mostly centers around 1) a new benchmark; 2) reward design; 3) better experimental results. The concerns are as follows:

1. Generalizability: The proposed methodology relies heavily on the execution of GeoGebra code. The inherent limitations in the expressiveness of this specific code representation raise valid concerns regarding the framework's adaptability. It is unclear how well this system's performance would generalize to broader multimodal tasks that lack a formal, domain-specific execution environment.

2. Presentation: The narrative structure, particularly in Section 3, leans heavily into "storytelling" and introduces an excess of formulas and terminology dwelling, which obscures the core technical methodology. Furthermore, the repeated framing of the model's behavior shift as an "Aha Moment" (e.g., in Sections 5.4 and 5.5) feels overstated and scientifically imprecise. The observed entropy changes primarily indicate a redistribution of token generation behavior rather than a cognitive breakthrough. Sections 5.4 and 5.5 essentially present standard qualitative and disentangled analyses showing expected post-RL performance gains resulting from the multi-faceted reward design. Rebranding these standard, albeit positive, improvements as an "Aha moment" is unnecessary and detracts from the professional, objective tone of the paper.

3. Novelty: While the tri-perspective reward design introduces a novel and tailored approach specifically to interleaved geometric reasoning, the resulting performance gains are largely unsurprising. These improvements align with expected outcomes demonstrated by a substantial body of recent work on RLVR. Because the success of this RL paradigm is well-established, the core methodological contribution feels somewhat incremental, raising valid concerns about its broader impact and applicability to domains beyond this paper.

---

> ### Author Rebuttal · Authors · 2026-03-31
>
> We thank Reviewer 2LYZ for the constructive feedback. We appreciate your recognition of the benchmark, the tri-perspective reward design, and the SFT failure analysis, and we are glad to address each concern below.
>
> ---
>
> **W1: Generalizability beyond GeoGebra**
>
> GeoGebra is not an ad-hoc choice but a widely adopted formal environment in geometry research. Recent works on automated proving and geometric verification build on GeoGebra or equivalent executable kernels, because geometry demands deterministic correctness checks that no surface-level metric can provide.
>
> More importantly, Faire's core principle is not tied to any specific DSL. Each component of the tri-perspective reward maps to a general abstraction: formal validity through any executable backend, perceptual quality through any VLM judge, and semantic alignment through any structured parser. Swapping GeoGebra for Python plotting or symbolic engines would only require replacing the execution layer while keeping the reward architecture intact.
>
> Empirically, the strongest evidence comes from transfer to external benchmarks outside our training pipeline. On GenExam, Faire achieves the best structural correctness (Math-Str 9.3) while matching the best relational score (Math-Rel 52.3), outperforming GPT-Image-1 (Table 3). On GGBench, Faire attains the highest overall interleaved score (VLM-I 63.13), surpassing GPT-5 (57.08) and Claude Sonnet 4.5 (54.11) at a much smaller scale (Table 7, Appendix). On MathVista, Faire reaches 81.8% versus 77.2% for its base model Qwen3-VL. These results across three independent benchmarks confirm that functional alignment generalizes beyond GeoGebra-specific training data.
>
> ---
>
> **W2: Presentation and the behavioral shift framing**
>
> We appreciate this critique and agree that the anthropomorphic language should be toned down. That said, we believe the underlying phenomenon itself is not a standard post-RL outcome and does merit a descriptive label. In typical RLVR settings, SFT provides a competent initialization that RL then refines incrementally. In our setting, SFT on interleaved data actively degrades performance relative to text-only SFT (62.48% vs. 68.13%), meaning the very data designed to teach interleaving becomes counterproductive. After RL, the same interleaved format reverses this degradation and surpasses the text-only ceiling (74.83% vs. 71.21%). This is a qualitative phase transition, not a gradual improvement, and we believe it warrants a concise descriptive term for ease of reference.
>
> In the revision, we will replace the anthropomorphic framing with more precise language, describing this as a transition from format imitation to functional grounding. We will also streamline the notation in Section 3 and reduce terminological overhead to improve accessibility.
>
> ---
>
> **W3: Novelty relative to standard RLVR**
>
> We respectfully argue that our contribution goes beyond applying RL to a new domain. The central finding is that interleaved multimodal reasoning presents a failure mode absent in unimodal RLVR, namely the SFT paradox described above. In text-only math reasoning, SFT provides a strong initialization and RL refines it incrementally. In interleaved geometry, SFT on multimodal traces causes negative transfer that RL must first reverse before any gains can be realized. This phenomenon has not been documented or analyzed in prior RLVR work.
>
> We provide two controlled experiments showing that standard RL paradigms cannot resolve this problem on their own.
>
> **Table R1. RL vs. Best-of-N on the SFT model.**
>
> | Method | Acc (%) | Drawing Score |
> |:---|:---:|:---:|
> | SFT (Interleaved) | 62.48 | 35.23 |
> | SFT BoN (N=5) | 66.14 | 36.18 |
> | SFT BoN (N=10) | 69.73 | 37.92 |
> | RL (Exec-only) | 64.29 | 35.65 |
> | Faire (Ours) | **74.83** | **45.37** |
>
> Even with N=10, Best-of-N selection (69.73%) falls well short of Faire (74.83%), and the drawing score gap is even larger (37.92 vs. 45.37). The RL (Exec-only) baseline, which uses only answer correctness and code executability as rewards, achieves just 64.29%, comparable to SFT BoN at N=5. This directly demonstrates that a standard RLVR objective is insufficient for interleaved multimodal tasks. Only our tri-perspective reward enables the model to internalize the causal coupling between construction and deduction, an outcome that no amount of sampling from the SFT distribution can achieve.
>
> Beyond geometry, the principle that generated intermediate states must be functionally grounded through causal verification applies broadly. Any domain where models interleave tool invocation with reasoning will face analogous alignment challenges, and our framework provides a concrete, transferable template.
>
> ---
>
> We hope these clarifications address the concerns about generalizability, presentation, and novelty. We have revised the manuscript accordingly and welcome any further discussion. Thank you again for the constructive feedback.

---

> > ### Author Rebuttal · Reviewer_2LYZ · 2026-04-03
> >
> > Thanks for the clarifications. My concerns are addressed and I look forward to seeing the modifications in the revised version. I have adjusted my scores accordingly.

---

### Official Review · Reviewer_Vsv7 · 2026-03-11

**Soundness:** 3
**Presentation:** 3
**Significance:** 3
**Originality:** 3
**Overall Recommendation:** 4
**Confidence:** 2

**Summary:**

This paper examines interleaved geometric reasoning, which requires models to alternate between textual deduction and executable diagram construction. The authors identify a surprising "SFT paradox" wherein supervised fine-tuning on interleaved traces actually degrades performance compared to text-only training. They argue that SFT induces superficial distributional alignment, failing to capture the causal dependency between diagrams and reasoning (T → V → Y). To address this, they propose Faire, an RL framework that enforces functional alignment through a tri-perspective verification system covering geometric/formal correctness (via executable assertions), perceptual usability (adjudicated by a VLM), and semantic alignment (between code and textual intent). Experiments on the proposed Faire-Bench demonstrate that RL reverses the SFT degradation, substantially improves both answer accuracy and functional metrics, and achieves strong correlation with automated verifier scores and human judgments.

**Compliance With Llm Reviewing Policy:**

Affirmed.

**Final Justification:**

I have read the authors' rebuttal and other reviews. My concerns have been fully addressed. I think the revised version should be accepted.

**Key Questions For Authors:**

See Weaknesses.

**Limitations:**

Yes.

**Strengths And Weaknesses:**

Strengths:
1. The distinction between distributional and functional alignment provides a theoretical framing for why naive interleaved SFT fails. The tri-verifier design is well-motivated by the causal mediation perspective (ingress/mediator/egress). The "Aha moment" behavioral analyses offer convincing evidence that the model internalizes plotting as a functional tool rather than decorative formatting.
2. The ablation studies effectively isolate the SFT paradox and demonstrate that all three verifiers are necessary for peak performance. The multi-dimensional evaluation (answer accuracy, tri-verifier scores, surface similarity, human correlation) convincingly shows that surface metrics (SSIM, LPIPS) are poor proxies for geometric correctness. Transfer results on GenExam further validate that gains reflect structural understanding rather than cosmetic improvements.
3. The work addresses a timely bottleneck in multimodal reasoning: making visual construction operationally useful. The executable GeoGebra pipeline and verifiable assertions provide a concrete recipe that could influence future RL-for-tool-use research, offering a principled alternative to purely aesthetic or text-similarity-based rewards.

Weaknesses:
1. The reliance on a VLM judge for perceptual rewards introduces potential reward hacking risks, yet no ablation tests evaluate alternative judges (e.g., Qwen-VL vs. GPT) to verify score stability. Furthermore, the semantic verifier lacks precision/recall metrics for the $\psi(T)$ and $\phi(c)$ extraction, making it unclear how errors in intent parsing might propagate during training.
2. The empirical results are impressive, but the baseline evaluations are somewhat ambiguous. It is unclear if the evaluated baseline models (e.g., GPT-4o, Gemini-2.5-Pro) were granted the same interactive, interleaved tool-execution loop (receiving rendered visual states back during generation) or if they were evaluated in a single-pass generation setting. To truly isolate the benefit of RL, it would also be valuable to see a test-time search baseline (e.g., Best-of-N sampling on the SFT model using your verifiers) to rule out the possibility that the gains are primarily driven by inference-time filtering rather than internalized functional alignment.

---

> ### Author Rebuttal · Authors · 2026-03-31
>
> We thank Reviewer Vsv7 for the thoughtful evaluation and for raising two concerns that go to the heart of whether Faire's gains are genuine. We address both in full below.
>
> ---
>
> **W1: Reward hacking risks and verifier stability across different judge models**
>
> The concern that a VLM-based reward might be exploited is well-founded in principle, but several empirical observations argue against it in our setting. The strong human-verifier correlation ($r=0.9578$, Figure 8) shows that automated score improvements translate faithfully into perceived diagram quality. Moreover, the large task-level gains (Acc from 62.48% to 74.83%, Drawing from 35.23 to 45.37) are accompanied by consistent improvements across all three verification axes rather than a spike in one, making it unlikely that the model is gaming any single signal.
>
> To directly test score stability, we evaluated two alternative judge models by measuring their agreement with human experts on all three verification perspectives.
>
> **Table R1. Human agreement (%) of different judge models.**
>
> | Judge Model | Alignment | Semantic | Geo Assert |
> |:---|:---:|:---:|:---:|
> | Faire (Ours) | **95.40** | **93.27** | **94.45** |
> | GPT-5.2 | 86.40 | 83.40 | 72.14 |
> | Qwen3-VL-32B | 76.40 | 64.60 | 67.14 |
>
> Our specialized verifier achieves substantially higher agreement with human judgments than either generalist model, especially on Geo Assert where formal geometric predicates are involved. Generalist models like Qwen3-VL-32B drop to 67.14% agreement on this axis, confirming that domain-specific verification is necessary for reliable reward signals. The high overall agreement rates also imply low false positive and negative rates. Although we did not run a full precision/recall analysis for the semantic parser's intent extraction, the 93.27% human agreement on the Semantic axis suggests that parsing errors do not meaningfully propagate into training signals. We will add these judge-comparison results to the revised manuscript.
>
> ---
>
> **W2: Baseline fairness and Best-of-N analysis**
>
> To clarify the evaluation protocol, all baseline models including GPT-4o and Gemini-2.5-Pro were evaluated under the exact same interleaved generation protocol as Faire. Each model alternated between reasoning and code generation within a single inference session, and the generated GeoGebra code was executed and rendered as part of the evaluation pipeline. No model received privileged access to rendered visual states during generation that others did not. We will make this protocol explicit in the revised experimental setup section to avoid any ambiguity.
>
> To isolate the contribution of RL from inference-time selection, we ran a Best-of-N (BoN) study on the SFT checkpoint using our tri-perspective verifiers as the selection criterion.
>
> **Table R2. RL vs. Best-of-N on the SFT model.**
>
> | Method | Acc (%) | Drawing Score |
> |:---|:---:|:---:|
> | SFT (Interleaved) | 62.48 | 35.23 |
> | SFT BoN (N=5) | 66.14 | 36.18 |
> | SFT BoN (N=10) | 69.73 | 37.92 |
> | RL (Exec-only) | 64.29 | 35.65 |
> | Faire (Ours) | **74.83** | **45.37** |
>
> Even at N=10, Best-of-N selection (69.73%) remains well below Faire (74.83%), and the drawing score gap is even wider (37.92 vs. 45.37). Notably, the BoN drawing score plateaus quickly while Faire's 45.37 represents a qualitative leap. This confirms that Faire's gains cannot be attributed to inference-time filtering alone. RL genuinely shifts the generation distribution toward more robust interleaved traces, producing better candidates on average rather than selecting lucky outliers from an unchanged distribution.
>
> The Exec-only RL baseline is equally informative. By rewarding only answer correctness and code executability, it reaches just 64.29%, comparable to SFT BoN at N=5. This shows that a naive RL objective is insufficient for interleaved tasks, because functional coupling between diagram construction and reasoning can only emerge when all three verification signals jointly shape the reward. Without the semantic and perceptual components, RL optimizes for superficially correct outputs that fail to ground the geometric state needed for reliable deduction.
>
> The SFT paradox itself reinforces this conclusion. Under SFT, the model systematically produces interleaved traces with misaligned diagrams, and no amount of reranking can fix a distribution whose construction-reasoning couplings are fundamentally broken. RL resolves this at training time, ensuring well-grounded traces by default rather than by chance. Taken together, these experiments confirm that Faire's gains reflect a genuine distributional shift toward functionally aligned interleaved reasoning, not an artifact of test-time selection.
>
> ---
>
> We have updated the manuscript with the alternative-judge comparison, the BoN analysis, and a clarified baseline protocol. Thank you again for raising these important points.

---

> > ### Author Rebuttal · Reviewer_Vsv7 · 2026-04-02
> >
> > I have read the authors' reply and other reviews. I will keep my positive score.

---

### Official Review · Reviewer_myov · 2026-03-13

**Soundness:** 3
**Presentation:** 3
**Significance:** 3
**Originality:** 3
**Overall Recommendation:** 5
**Confidence:** 4

**Summary:**

This paper identifies the SFT paradox in geometric interleaved reasoning—SFT on interleaved plot-solution data degrades performance by inducing distributional alignment without functional understanding. Faire addresses this via RL (GRPO) with a tri-perspective verifier (geometric consistency, perceptual admissibility, semantic alignment), achieving strong verification scores on the proposed Faire-Bench.

**Compliance With Llm Reviewing Policy:**

Affirmed.

**Key Questions For Authors:**

1. Does the SFT paradox persist with larger models (e.g., 72B)? Does scale mitigate distributional alignment issues?
2. What fraction of Faire-Bench problems can be solved by text-only reasoning without any plotting? This would validate whether the interleaved format is necessary.
3. What is the computational overhead of tri-perspective verification during RL training?
4. How sensitive is performance to the perceptual threshold and reward weighting? What are the verifier false positive/negative rates?
5. How does Faire perform on established geometry benchmarks where interleaved plotting may not be required?

**Limitations:**

The paper includes an impact statement but no explicit limitations section. Key unaddressed limitations: (1) single-model evaluation, (2) evaluation only on self-constructed benchmarks, (3) GeoGebra-specific verification limits generalization, (4) no text-only baseline to validate necessity of interleaved format, (5) no analysis of RL training cost.

**Strengths And Weaknesses:**

**Strengths:**

1. The SFT paradox is clearly demonstrated and the distinction between distributional and functional alignment is insightful and well-formalized in Section 3.2, with clean information-theoretic arguments.

2. The tri-perspective verifier covers complementary aspects of interleaved generation quality. The leave-one-out ablation (Table 4) confirms each component is necessary and no single proxy suffices.

3. Faire substantially outperforms GPT-5.2 on verification scores (45.37 vs 24.62). Human correlation r=0.9578 validates the automated metrics. GenExam results (Table 3) show structural transfer beyond geometry.

4. The dataset construction pipeline with three complementary filters (alignment, semantic, geo-assert) is rigorous. Multi-image coverage at 38-44% is a distinctive and valuable feature.

**Weaknesses:**

1. All training uses only Qwen3-VL-8B. It is unknown whether the SFT paradox persists at larger scales or with different architectures where models may already internalize functional alignment.

2. Evaluation is exclusively on Faire-Bench and GenExam, both self-constructed. No established geometry benchmarks (GeoQA, MathVista, Geometry3K) are included, making generalizability hard to assess.

3. The "Aha moment" is central to the narrative but lacks formal definition. The evidence relies on post-hoc entropy analysis (Sec 5.3) and qualitative examples (Fig 6-7), with no quantitative criterion for when it has occurred. Recent work on emergent interleaved reasoning properties (e.g., autonomous mode switching, unseen visual manipulations) provides more rigorous behavioral definitions that this paper could learn from.

4. The paper assumes interleaved reasoning is always desirable for geometric tasks but does not verify this. No experiment tests whether these problems can be solved via text-only reasoning, which would establish that the interleaved format is truly necessary rather than just one viable approach.

5. No comparison with other RL reward designs (DPO, RLHF variants) or tool-augmented reasoning approaches. It is unclear how much of the gain comes from RL itself versus the specific tri-perspective reward.

6. The tri-perspective verifier depends on GeoGebra executability and a VLM judge. No analysis of verifier reliability (false positive/negative rates) or sensitivity to the threshold and weighting is provided.

---

> ### Author Rebuttal · Authors · 2026-03-31
>
> We sincerely thank Reviewer myov for the thorough evaluation. Your recognition of the SFT paradox, the tri-perspective verifier, and the dataset pipeline is encouraging, and your suggestions have prompted several valuable additions.
>
> ---
>
> **W1 & Q1: Does the SFT paradox persist at larger model scales?**
>
> We ran new experiments on Qwen3-VL-32B, the largest dense model in the family.
>
> **Table R1. SFT paradox across model scales.**
>
> | Paradigm | Size | SFT Acc (%) | SFT Draw | RL Acc (%) | RL Draw |
> |:---|:---|:---:|:---:|:---:|:---:|
> | Text-only | 8B | 68.13 | - | 71.21 | - |
> | Interleaved | 8B | 62.48 | 35.23 | 74.83 | 45.37 |
> | Text-only | 32B | 74.21 | - | 78.12 | - |
> | Interleaved | 32B | 71.28 | 46.24 | 80.63 | 56.23 |
>
> At 32B, interleaved SFT (71.28%) still falls short of text-only SFT (74.21%), while RL recovers and exceeds this ceiling (80.63%). The paradox therefore stems from a training-objective mismatch rather than insufficient model capacity. We plan to extend to MoE architectures in the final version.
>
> ---
>
> **W2 & Q5: Evaluation on established external benchmarks**
>
> We would like to clarify that GenExam and GGBench are independently published external benchmarks, not part of our construction. Following your suggestion, we additionally evaluated Faire on MathVista, a widely used multimodal math benchmark.
>
> **Table R2. Results on MathVista.**
>
> | Model | MathVista Acc (%) |
> |:---|:---:|
> | GPT-4o | 63.8 |
> | Qwen2.5-VL-7B | 68.1 |
> | Qwen3-VL-7B | 77.2 |
> | Faire (Ours) | **81.8** |
>
> Faire outperforms all baselines by a clear margin. Together with the GenExam and GGBench results already reported in the paper, this provides convergent evidence of generalizability across three independent external benchmarks.
>
> ---
>
> **W3: Quantitative characterization of the behavioral shift**
>
> We agree that entropy analysis alone is suggestive rather than definitive. To offer a sharper quantitative criterion, we computed a token-level log-probability shift $\Delta\log p = \log P_{\text{RL}} - \log P_{\text{SFT}}$ for every token and aggregated by semantic category, following recent work on RLVR update directions [1]:
>
> - Reasoning tokens: mean $\Delta\log p = -0.0117$, with 27.7% receiving a positive shift.
> - Plotting tokens: mean $\Delta\log p = +0.3771$, with 26.4% receiving a positive shift.
>
> The sparse yet high-magnitude positive shift concentrates on plotting tokens, indicating that RL selectively reinforces trajectories where drawing genuinely supports deduction. Reasoning logic remains stable (mean near zero), confirming that RL enhances tool use without disrupting the deductive chain. We will incorporate this analysis in the revision.
>
> [1] Kexin Huang, et al. On the Direction of RLVR Updates for LLM Reasoning: Identification and Exploitation, ICLR 2026.
>
> ---
>
> **W4 & Q2: Is the interleaved format truly necessary?**
>
> Table 5 in the paper provides a controlled comparison. Under SFT, interleaved training (62.48%) actually hurts accuracy relative to text-only training (68.13%), which might lead one to conclude that plotting is unnecessary. Yet after RL enforces functional alignment, the interleaved format (74.83%) substantially surpasses the text-only RL baseline (71.21%). The 3.62-point gap shows that properly grounded diagrams provide information that pure textual reasoning cannot recover on its own, an advantage most pronounced in categories like triangle congruence and function graphs where spatial configurations are ambiguous without explicit construction (Figure 1).
>
> ---
>
> **W5, W6 & Q4: Reward design choices and verifier reliability**
>
> Our leave-one-out ablation (Table 4) demonstrates that removing any single verifier consistently degrades both accuracy and drawing score. For instance, dropping the Geo Assert verifier reduces accuracy from 74.83% to 65.88%, and dropping Semantic reduces it to 66.24%. An execution-only reward yields only 64.29%, barely above the SFT baseline of 62.48%. These results confirm that the three signals are genuinely complementary and that no single proxy is sufficient.
>
> Regarding verifier reliability, human agreement rates reach 95.4% for Alignment, 93.3% for Semantic, and 94.5% for Geo Assert. The strong overall human-verifier correlation ($r=0.9578$, Figure 8) further validates that false positive and negative rates remain low. We adopt equal weighting because the three verifiers address mutually exclusive failure modes.
>
> ---
>
> **Q3: Computational overhead of tri-perspective verification**
>
> We deployed the three verifiers in parallel on independently hosted rendering services. During RL training, verification accounts for approximately 12.9% of total wall-clock time, with per-sample latencies of 0.68s (Alignment), 0.89s (Semantic), and 0.47s (Geo Assert). These costs are modest compared to the sampling phase and do not bottleneck training throughput.
>
> We have revised the manuscript to incorporate the above additions and look forward to any further questions.

---

> > ### Author Rebuttal · Reviewer_myov · 2026-04-07
> >
> > The rebuttal addresses my questions well. Please include the new experimental results in the revision. I am maintaining my score of Accept.

---

### Decision · Program_Chairs · 2026-04-30

**Decision:**

Accept (spotlight)

**Comment:**

This paper identifies a phenomenon called the SFT paradox that SFT on interleaved plot-solution data actually hurts geometric reasoning performance compared to text-only baselines. The authors argue that SFT induces distributional alignment without functional understanding, meaning the model learns to reproduce the format of interleaved plotting but never internalizes the causal relationship between the diagram and the reasoning steps. To fix this, they propose an RL framework with a tri-perspective verifier covering geometric correctness, perceptual usability, and semantic alignment, which enforces what they call functional alignment. The paper also introduces a new benchmark with GeoGebra script annotations. The reviews were generally positive. Key concerns included whether gains could be explained by inference-time filtering rather than genuine learning, potential reward hacking from the VLM-based perceptual judge, generalizability beyond GeoGebra, novelty relative to standard RLVR, and evaluation only on self-constructed benchmarks. The authors handled the rebuttal well. The most important piece of new evidence was the Best-of-N analysis, even with N=10 sampling from the SFT model and selecting with the same tri-perspective verifiers, the result still falls well below Faire, and the drawing score gap is even wider. This rules out the possibility that the gains are just from inference-time filtering.